# Fingerprints of Super Resolution Networks

**Jeremy Vonderfecht**                                    *vonder2@pdx.edu*
*Department of Computer Science*
*Portland State University*

**Feng Liu**                                              *fliu@pdx.com*
*Department of Computer Science*
*Portland State University*

**Reviewed on OpenReview:** *https: // openreview. net/ forum? id= Jj0qSbtwdb*

## Abstract

Several recent studies have demonstrated that deep-learning based image generation models, such as GANs, can be uniquely identified, and possibly even reverse-engineered, by the fingerprints they leave on their output images. We extend this research to single image super-resolution (SISR) networks. Compared to previously studied models, SISR networks are a uniquely challenging class of image generation model from which to extract and analyze fingerprints, as they can often generate images that closely match the corresponding ground truth and thus likely leave little flexibility to embed signatures. We take SISR models as examples to investigate if the findings from the previous work on fingerprints of GAN-based networks are valid for general image generation models. We show that SISR networks with a high upscaling factor or trained using adversarial loss leave highly distinctive fingerprints, and that under certain conditions, some SISR network hyperparameters can be reverse-engineered from these fingerprints.

## 1 Introduction

Recent progress in deep-learning based image synthesis has dramatically reduced the effort needed to produce realistic but fake images (Tolosana et al., 2020). But just as a criminal may leave fingerprints at a crime scene, image synthesis networks leave telltale "fingerprints" on the images they generate (Marra et al., 2019).

Researchers have sought to extract increasingly detailed information about image synthesis networks from these fingerprints. A popular form of this problem is *deepfake detection* (Dolhansky et al., 2019), which seeks to extract a single bit of information: is a particular image real or fake? Going further, *model attribution* seeks to identify the particular image generation model that produced an image (Yu et al., 2019). *Model parsing* goes even further, seeking to infer design details of the image generation model (Asnani et al., 2021).

Model fingerprints have been studied primarily to identify and track down sources of misinformation. Therefore, the types of models used for deepfakes, such as generative adversarial networks (GANs), have received the most attention. The tasks performed by such models are open-ended: they permit many possible valid outputs. For example, unconditional GANs may generate any image in the training domain. We hypothesize that this open-endedness is a key requirement for distinctive model fingerprints. To test this hypothesis, we are interested in studying model fingerprinting problems, such as attribution and parsing, for a less open-ended image synthesis task. Single-image super-resolution (SISR) is a good choice of task for two reasons. First, SISR is a highly active area of research, with many state-of-the-art models available online for free. Second, a SISR network's output is highly constrained by its input, and often hews closely to some optimal solution. For example, different $L_1$-optimized SISR models are known to converge on super-resolved outputs which are visually very similar (Sajjadi et al., 2017). If our hypothesis about fingerprints is correct, such SISR models will be much harder to differentiate from each other.

To understand the fingerprints of SISR networks, we collect photographs from Flickr and super-resolve each of them with 205 different SISR models. These 205 models consist of 25 pretrained models published online by other researchers, and 180 models which we have trained ourselves by systematically varying four experimental hyperparameters: architecture, super-resolution scale, training dataset, and loss function. We then train an extensive collection of image classifiers to perform model attribution, and to predict the values of our experimental hyperparameters. By systematically reserving different subsets of the SISR models for testing, we investigate how our model attribution and parsing classifiers generalize. Our contributions are as follows:

- We develop a novel dataset of 205,000 super-resolved images generated by 205 different SISR models, all of which will be made publicly available.

- We analyze the factors that contribute to the distinctiveness of an SISR model fingerprint. We show that the choice of scaling factor and loss function significantly impacts distinctiveness, corroborating our hypothesis that more open-ended training objectives lead to more distinctive fingerprints.

- As Yu et al. (2019) showed for GANs, we show that the fingerprints of SISR models trained with an adversarial loss are highly sensitive to small changes in hyperparameters, such as random seed.

- We study the generalization of our SISR model attribution classifier to models outside the training set. We show that our attribution classifier generalizes well from our contrived training set to real-world models, with architectures and loss functions not seen during training.

- We train a set of model parsing classifiers to predict the hyperparameters of the SISR models. We show that under certain conditions, it is possible to reverse-engineer some of a model's hyperparameters from its output images.

## 2 Related Work

**Single image super-resolution:** Recent years have seen rapid progress in deep-learning based SISR models. These days, there are a profusion of such methods available. We choose SISR models as our subject of study for this reason. A diverse set of state-of-the-art SISR models form the foundation of our experiments. We select a collection of SISR models presented in recent papers based on their reproducibility and their high-quality results (Chen et al., 2021; Dai et al., 2019; Guo et al., 2020; Kim & Son, 2021; Li et al., 2019; Liang et al., 2021; Lim et al., 2017; Ma et al., 2020; Mei et al., 2021) (Sajjadi et al., 2017; Wang et al., 2021; 2018b;c; Zhang et al., 2018a;b). Our 180 custom-trained SISR models were trained using the BasicSR framework (Wang et al., 2018a).

**Model attribution:** To identify the source of synthetic images, model attribution methods can look either for watermarks (signatures deliberately encoded into each output image by the network author) (Adi et al., 2018; Hayes et al., 2020; Skripniuk et al., 2020; Yu et al., 2021; Zhang et al., 2020), or for unintentional statistical anomalies in the generated images which are unique to a particular model, which we call "fingerprints". We focus on detecting these fingerprints, which can appear without deliberate intervention by the network author.

There is no consensus on a precise definition of the term "model fingerprint". Other works have defined fingerprints in terms of some specific image feature extraction technique. Marra et al. (2019) formulate model fingerprints as the average noise residual across many of the model's output images. In Yu et al. (2019) and (Asnani et al., 2021), fingerprints are feature vectors encoded by deep image classifiers trained to do model attribution/parsing. For our purposes, we use the term "fingerprint" more generally, to refer to all statistical irregularities in an image which reveal information about the generating model.

Generative adversarial networks have been shown to possess uniquely identifying fingerprints (Marra et al., 2019; Yu et al., 2019). These fingerprints have been extracted with convolutional networks (Xuan et al., 2019; Yu et al., 2019; Jain et al., 2021), and with hand-crafted features (Goebel et al., 2020; Guarnera et al., 2020; Marra et al., 2019). Other works have shown that GAN inversion can be a useful way to attribute

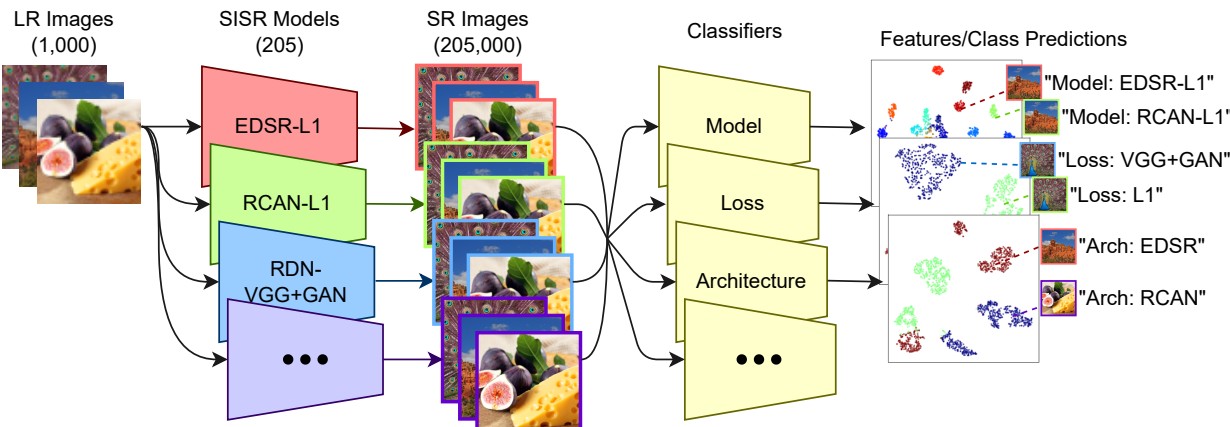

Figure 1: An overview of our experimental setup. 1,000 low-resolution images are fed through each of 205 SISR models to produce a dataset of 205,000 super-resolved images. We then train our model parsing and attribution classifiers on this dataset of super-resolved images.

images to GANs (Albright & McCloskey, 2019; Zhang et al., 2021). All of these methods focus on finding the fingerprints of GANs or Variational Auto Encoders (VAEs). However, the relevance of these results to image enhancement models, such as SISR networks, is *a priori* unclear. We present the first study of fingerprints for SISR models, and show that SISR models also leave unique fingerprints which are identifiable by CNNs.

**Reverse-engineering/model parsing**: We are not the first to attempt to reverse-engineer the hyperparameters of black-box neural networks. (Oh et al., 2019) reverse-engineer many fine-grained architectural choices of small image classification networks . However, their approach requires that the reverse-engineer can feed their own specially designed inputs into the black-box network and observe the resulting output. By contrast, we attempt to infer the model hyperparameters from an arbitrary output image.

In a concurrent work, Asnani et al. (2021) train a convolutional network to extract a fingerprint from a generated image, and to predict the hyperparameters of various image generation models from this fingerprint. Their method, like ours, can be used for attribution and model parsing. Their study covers a diverse domain of 116 image generation models, including GANs, variational autoencoders, and two SISR networks: ESRGAN (Wang et al., 2018b) and SRFLOW (Lugmayr et al., 2020). Our work, by contrast, is focused specifically on SISR models, which generate images very close to a ground truth and likely leave less flexibility to embed fingerprints. We are the first to study this hyperparameter reverse-enegineering problem specifically for SISR models.

## 3 Study Setup

We investigate what choices of SISR model hyperparameters lead to distinctive model fingerprints, and which model hyperparameters can be reverse-engineered using model parsing. To explore these questions carefully, we need a dataset of SISR models that varies the hyperparameters we wish to experiment on, while holding all other hyperparameters constant. We choose 5 SISR model hyperparameters to analyze: model architecture, super-resolution scale factor, loss function, training dataset, and random seed. We train 180 SISR models with various combinations of these experimental hyperparameters, and add in 25 more SISR models from previous works for additional diversity. We construct our main dataset by super-resolving 1,000 images by each of our 205 SISR models. We use this dataset to train several image classifiers for model attribution and parsing tasks. Finally, we analyze the performance of these classifiers. Figure 1 shows an overview of our process. Further details about about our setup are available in the supplementary material.

We want our collection of SISR models to meet the following criteria:

1. **Realistic**: Models are representative of those in recent literature; they are not contrived toy models.

2. **Diverse**: Our models should span many architectures, loss functions, and training sets.

3. **Large**: We want a large number of SISR models, so that the model classifiers can begin to generalize across the SISR model space.

4. **Uniform**: The hyperparameters of our SISR models should be uniformly distributed and independent from each other to prevent spurious correlations that could confound our analysis.

### 3.1 Single Image Super-Resolution Models

To make our model collection realistic and diverse, we include 25 real-world pretrained super-resolution models, published between 2017 and 2021 (See Table 1). Unfortunately, there are not enough pretrained SISR models available online to make our dataset very large. But even more importantly, distribution of hyperparameter values among these 25 SISR models is neither uniform nor independent. For example, seven out of the eight 2X scale pretrained models are $L_1$-optimized. Such correlations are confounding factors in our analysis, and we would prefer to avoid them.

In the most common setup, SISR models are determined by four key factors: network architecture, training dataset, super-resolution scale, and loss function. All hyperparameters of SISR model training and inference are determined by these factors, with the exception of optimizer-based hyperparameters such as learning rate, batch size, etc. We were interested in studying the impact of each of these key factors on model fingerprints. So we built our collection of custom-trained SISR models from different choices of these four key factors, which we will refer to as *experimental hyperparameters*. We add a fifth experimental hyperparameter, random training seed, as a kind of control group, under the assumption that changing any meaningful parameter of the SISR model will have an impact at least as significant as changing the random seed. Our experimental hyperparameters and their values are as follows:

Table 1: Pretrained SISR models we use.

| Name | Loss | Scale(s) |
|------|------|----------|
| EDSR | $L_1$ | 2X, 4X |
| EnhanceNet | Adv. | 4X |
| RDN | $L_1$ | 2X, 4X |
| ProSR | $L_1$ | 4X |
| ProGanSR | Adv. | 4X |
| RCAN | $L_1$ | 2X, 4X |
| ESRGAN | Adv. | 4X |
| SAN | $L_1$ | 4X |
| SRFBN | $L_1$ | 2X, 4X |
| SPSR | Adv. | 4X |
| DRN | $L_1$ | 4X |
| NCSR | Adv. | 4X |
| SwinIR [1] | $L_1$, Adv. | 2X, 4X |
| LIIF | $L_1$ | 2X, 4X |
| Real-ESRGAN | Adv. | 2X, 4X |
| NLSN | $L_1$ | 2X, 4X |

[1] There are 3 pretrained SwinIR models: $(L_1, 2X)$, $(L_1, 4X)$, and (Adv, 4X)

1. **Architecture**: The choices are **EDSR** (Lim et al., 2017), **RDN** (Zhang et al., 2018b), **RCAN** (Zhang et al., 2018a), **NLSN** (Mei et al., 2021), and **SwinIR** (Liang et al., 2021).

2. **Dataset**: The super-resolution dataset used for training the model. The choices are **DIV2K** (Agustsson & Timofte, 2017) or **Flickr2K**, originally collected by Lim et al. (2017). To see if using a smaller training dataset might lead to a more distinctive model fingerprint, we also trained SISR models with just one quarter of the total training data available from these two datasets, effectively creating two more dataset choices, $\frac{1}{4}$**DIV2K** and $\frac{1}{4}$**Flickr2K**

3. **Scale**: Scaling factor by which to upsample the low-resolution input image; either **2X** or **4X**. To be clear, this is the scaling factor for the linear dimension of the image, not the total number of pixels; a 2X-upsampled image has four times as many pixels.

   **Loss**: Loss function to optimize during training. Choices are the $L_1$ **norm** (which is standard in the super-resolution literature), **VGG+adv.** loss, or **ResNet+adv.** loss. VGG+adv. loss is the same linear combination of VGG-based perceptual loss and adversarial loss that was used in SRGAN (Wang et al., 2018b):

$$l_{VGG+adv} = l_{percep/\phi} + 10^{-3}l_{adv}$$
$$l_{percep/\phi} = MSE(\phi(I^{HR}), \phi(I^{SR}))$$
$$l_{adv} = -\log D(I^{SR})$$

Where $I^{HR}$ is the high-resolution ground truth image, and $I^{SR}$ is the SISR model's output image, $\phi$ is the perceptual feature extractor, and $D$ is the discriminator network. The adversarial loss term $l_{adv}$ is standard for GANs: the negative log likelihood of the discriminator $D$'s probability that $I^{SR}$ is a natural image. The perceptual loss $l_{percep/\phi}$ measures the mean squared error between two feature embeddings of $I^{SR}$ and $I^{HR}$, from a pre-trained classification network $\phi$. For VGG+adv. loss, $\phi$ outputs internal activations from a pre-trained VGG net (Simonyan & Zisserman, 2014). ResNet+adv. loss is the same, except it uses a pretrained ResNet instead (He et al., 2016).

4. **Seed**: Yu et al. (2019) found that changing the random seed used for network initialization is sufficient to produce a GAN with a distinct fingerprint. To determine the effect of random seeds in our setting, we train copies of our models with three different seeds: 1, 2, or 3.

In total, there are 360 possible SISR models that could be trained from different combinations of these hyperparameters. To save time and computation, we only train the subset of these models whose random seed is 1 *or* whose training dataset is DIV2K. (We chose to limit these two parameters because we found that they had the smallest influence on the learned model.) This leaves us with 180 custom-trained SISR models.

## 3.2   Image Datasets

As discussed in Section 3.1, We employ two existing super-resolution image datasets, DIV2K and Flickr2K, to train our SISR models. We also create our own dataset of super-resolved images which we use to train the model attribution and parsing classifiers.

Our new image dataset consists of 1,000 photographs from Flickr. We query Flickr for 200 images from each of the following 5 image tags: food, architecture, people, animals, and landscapes. We select only images with at least two megapixel resolution. At full resolution, many images contain visible JPEG artifacts, so we downsample (Gaussian blur followed by bicubic downsampling) them to 960 pixels in their largest dimension, at which point any jpeg artifacts are imperceptible to us. We refer to this collection of images as the "Flickr1K dataset". Our final dataset comprises images from over 500 Flickr users and over 300 types of cameras, as measured by the metadata provided by the Flickr API.

To generate the final super-resolution dataset upon which we train our model attribution and parsing classifiers, we super-resolve each image in our Flickr1K dataset by each of our 205 SISR models. For each SISR model, for each Flickr image, we first downsample the image by the model's scaling factor using bicubic interpolation, and then super-resolve the downsampled image using the model. This gives us a dataset of 205,000 super-resolved images. Figure 2 shows several super-resolution examples.

## 3.3   Classification Networks

We train an extensive collection of model classification networks to perform model attribution and parsing. First, we needed to select an image classification network architecture. Wang et al. (2020) found that a ResNet50-based classifier (He et al., 2016) was capable of distinguishing CNN-generated images from natural images with surprisingly good generalization. Rössler et al. (2019) used XceptionNet (Chollet, 2017) to effectively detect deepfakes in the FaceForensics++ dataset. We also tried two additional state-of-the-art image classification networks: EfficientNet B2 (Tan & Le, 2019) and ConvNext (Liu et al., 2022). We chose the network with the highest accuracy on the 205-way model attribution problem, *i.e.*, "which of these 205 SISR models generated this image?" As shown in Table 2, we found that ConvNext achieved the highest accuracy.

Table 2: Accuracy (%) on 205-class model attribution.

| Architecture | Acc. |
| --- | --- |
| ConvNext | 94.3 |
| XceptionNet | 93.2 |
| Efficientnet | 92.9 |
| ResNet50 | 81.8 |

To adapt a ConvNext network pretrained on ImageNet to our classification problems, we replace the final fully-connected layer of the network with a randomly initialized one of the appropriate shape to output the right number of classes. The network is trained with cross-entropy loss. As in Rössler et al. (2019), we train just the final layer for three epochs, then we unfreeze all network weights and fine-tune the entire network

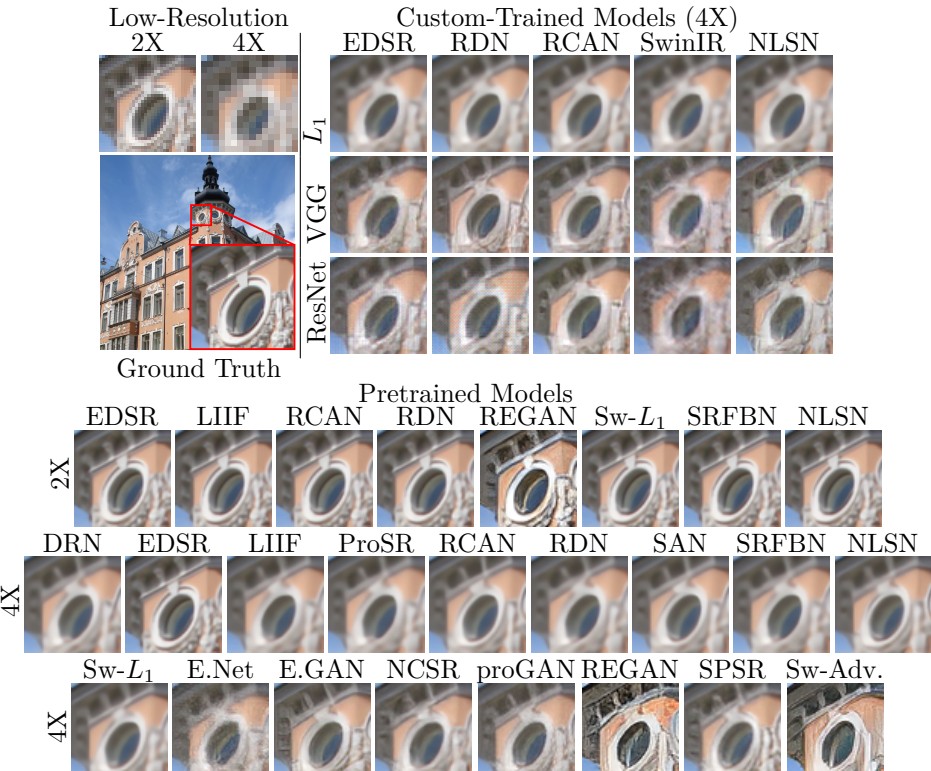

Figure 2: A small image patch super-resolved by a sample of the SISR models in our dataset. Abbreviations: REGAN: Real-ESRGAN, Sw-$L_1$: L1-optimized SwinIR, Sw-Adv: adversarially-optimized SwinIR, E.Net: EnhanceNet, E.GAN: ESRGAN. Best viewed zoomed in.

for 15 epochs. We use the Adamax optimizer with a learning rate of 0.0005, Batch size of 16. Our classifiers are trained with our super-resolved image dataset on just 800 images from each SISR model. We reserve an additional 100 images for validation, and 100 for testing. Images are cropped down to the network's input size. In the training set, images are randomly cropped. In validation and testing, they are center-cropped. All analyses presented in Section 4 are computed from this test set of 100 images per SISR model (205,00 images in total).

To account for random variation in our results, we train each classifier three times, starting from three different random seeds. Throughout Section 4, we report the mean and standard deviation of our accuracy scores across these three random seeds.

## 4 Experiments

Our experiments are organized around the analysis of two different problems: model attribution (Section 4.1) and model parsing (Section 4.2). We formulate both as classification problems, and train ConvNext models to solve them as described in Section 3.3.

Except where otherwise stated, all accuracy scores are reported as a *{mean}* ± *{standard deviation}* across three different classifiers, initialized with different random seeds but otherwise trained identically. T-SNE-based feature visualizations (Figures 3 and 5) are generated from just one of these three classifiers.

Table 3: Accuracy (%) of our custom model attribution classifier grouped by different hyperparameters. We report the average and std. of classification accuracies across three versions of each classifier (trained with three different random seeds but otherwise trained identically). For example, the average accuracy of our classifier on SISR models whose *scale* is *2X* is 95.4%.

| scale | accuracy | loss | accuracy | architecture | accuracy | dataset | accuracy |
|---|---|---|---|---|---|---|---|
| 2X | 95.4±0.7 | $L_1$ | 89.2±1.5 | DIV2K | 96.1±1.0 | EDSR | 94.9±1.3 |
| 4X | 96.6±0.4 | VGG+adv. | 99.2±0.2 | Flickr2K | 96.6±0.1 | RDN | 96.3±0.5 |
| | | Resnet+adv. | 99.6±0.1 | $\frac{1}{4}$ DIV2K | 94.9±0.5 | RCAN | 94.7±1.1 |
| | | | | $\frac{1}{4}$ Flickr2K | 96.3±0.4 | SwinIR | 98.0±0.4 |
| | | | | | | NLSN | 96.1±0.9 |

## 4.1 Model Attribution

How reliably can an SISR model be uniquely identified by its output images? What combinations of hyperparameters lead to distinctive fingerprints? To answer these questions, we train and analyze two attribution classifiers: the *custom model attribution classifier*, which is trained to distinguish between the 180 custom-trained SISR models, and the *pretrained model attribution classifier*, which is trained to distinguish between the 25 pretrained models. We discuss the comparative benefits and drawbacks of these two subsets of models in Section 3.1. Essentially, the custom models are a larger and more controlled sample, while the pretrained models are more realistic and diverse. (For the rest of the paper, we use "model" as shorthand for "SISR model".)

### 4.1.1 When are SISR Model Fingerprints Distinctive?

Do certain hyperparameter choices make SISR model finger-prints more or less distinctive? Super-resolution seeks to invert an image downscaling function whose input domain (high resolution images) is larger than its output range (low-resolution images). This means that each super-resolution input corresponds to many possible valid outputs. If the number of possible valid outputs is small, it is more likely that two different SISR networks will converge upon the same output. Therefore, we hypothesize that the distinctiveness of a super-resolution method increases with the size of this space of possible outputs.

From this hypothesis, it easily follows that 4X upscaling should produce more distinctive fingerprints than 2X. As discussed in Sajjadi et al. (2017), SISR models trained with an adversarial loss are incentivized to sample from this space of possible output images, while $L_2$-optimized networks are incentivized to aggregate over this space with a pixel-wise mean, effectively reducing the number of valid outputs. Sajjadi *et al.*observe that different $L_2$-optimized SISR models tend to converge upon the same unnaturally smooth super-resolved images, while SR images from adversarially-trained models are more diverse. We can expect $L_1$ loss to behave similarly, except it will aggregate over the space of possible images with a pixel-wise median instead of a mean.

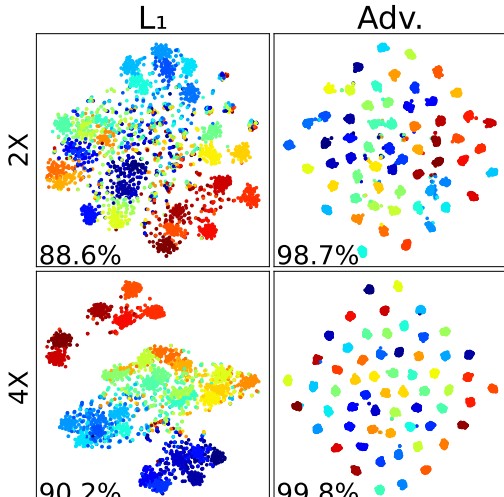

Figure 3: T-SNE visualizations of super-resolved image feature embeddings, grouped by scale and loss. Each point represents an image from the test set, colored according to the SISR model that generated it. Accuracies for each group (for this classifier) are in the lower left.

Therefore, we hypothesize that 4X SISR models leave more distinctive fingerprints than 2X, and adversarially-trained models leave more distinctive fingerprints than $L_1$ models.

To test this hypothesis, we examine the accuracy of our 180-class custom model attribution classifier grouped by each hyperparameter, as shown in Table 3. Classification accuracy varies significantly by scale and loss

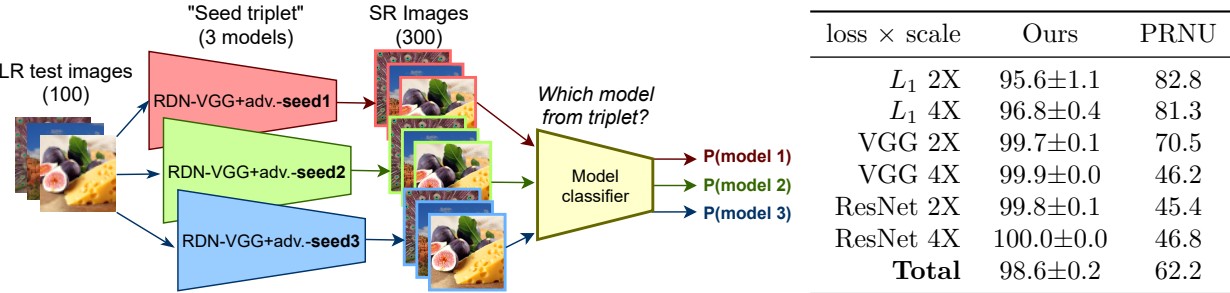

| loss $\times$ scale | Ours | PRNU |
|---|---|---|
| $L_1$ 2X | 95.6±1.1 | 82.8 |
| $L_1$ 4X | 96.8±0.4 | 81.3 |
| VGG 2X | 99.7±0.1 | 70.5 |
| VGG 4X | 99.9±0.0 | 46.2 |
| ResNet 2X | 99.8±0.1 | 45.4 |
| ResNet 4X | 100.0±0.0 | 46.8 |
| **Total** | 98.6±0.2 | 62.2 |

Figure 4 & Table 4: Distinguishing between models which differ only by seed. The table shows the accuracy (%) our custom model attribution classifier vs the PRNU-based classifier from Marra (2017). One of the 30 seed triplets, the (4X, $L_1$, NLSN) triplet, had a relatively low distinction accuracy of 87.3%. So the table is computed with NLSN models omitted.

function: average classification accuracy for 4X SISR models is 1.2% higher than for 2X models, and average classification accuracy for adversarially-trained models is 10.2% higher than for $L_1$ models.

Figure 3 shows a T-SNE embedding of the super-resolved image features disaggregated by scale and loss. We define *image features* as the 1024-dimensional vector of activations from the last layer of an attribution classifier. Class separation is better for 4X SISR models than 2X, and better for adversarial than $L_1$ models. This data supports our hypothesis that adversarial loss functions and higher super-resolution scales lead to more distinctive model fingerprints.

**Distinguishing Among Random Seeds:**

Yu et al. (2019) show that small variations in a GAN's hyperparameters can lead to highly distinctive GAN fingerprints. To test if this finding holds for SISR models, we evaluate the accuracy of our custom model classifier at distinguishing between groups of models which differ only by their random seed (as shown in Figure 4). Our custom-trained SISR model dataset contains 30 "seed triplets": sets of three SISR models which are identical except for their seed. Our custom-trained model classifier can distinguish between models in each triplet with an average of 98.6% accuracy. We interpret this as a confirmation that Yu *et al.*'s finding extends to this new domain.

For a baseline comparison, we evaluate the PRNU-based model attribution scheme from Marra et al. (2019) on this seed distinction problem. Marra *et al.*develop a simple scheme for image attribution based on handcrafted features. They denoise each image using BM3D (Dabov et al., 2007), and subtract that denoised image from the original image to obtain a "noise residual". The noise residuals of all training images from the same source are averaged together into a "fingerprint" for that image source. To perform attribution, they find the noise residual of the image in question, and attribute it to the source with the closest fingerprint, by euclidean distance. We find that our method achieves significantly better accuracy on this problem, which is significantly harder for the PRNU-based attribution scheme than the GAN attribution problems from (Marra et al., 2019). In an interesting reversal of a trend for our ConvNext classifier, $L_1$-optimized SISR models appear to leave more distinctive PRNU fingerprints than adversarially-optimized models.

### 4.1.2 Pretrained Model Attribution

Our 25 pretrained models have a greater diversity of architectures, loss functions, and datasets, and may be more representative of the kinds of SISR models encountered in real, practical use. So do our attribution results still hold for them? To test this, we train an attribution classifier to predict which of the 25 pretrained models produced a given super-resolved image. We find that the performance of this classifier improves significantly if we initialize it with weights from the custom model attribution classifier, instead of starting from the ConvNext model used for ImageNet classification.

Overall test-set attribution accuracy for these 25 pretrained models is 87.5±0.6%. Our hypothesis about scale and loss function still holds: accuracy among the (2X, $L_1$) models is 77.4±2.2%, while accuracy among

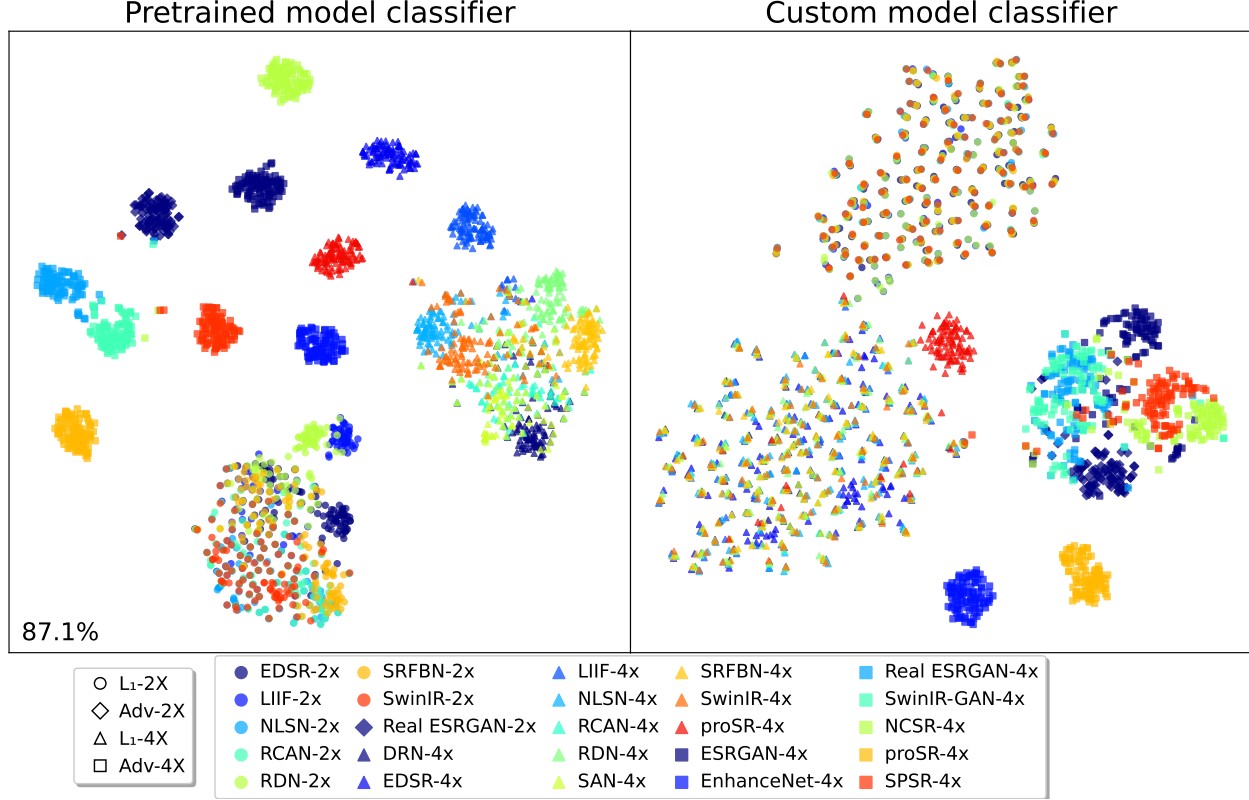

Figure 5: T-SNE Feature embeddings for images generated by our pretrained SISR models. Left: embeddings as encoded by one of the pretrained model attribution classifiers (attribution accuracy in lower-left). Right: embeddings by one of the custom model attribution classifiers, which was not trained on these models.

the $(4X, L_1)$ is $87.9\pm1.0\%$. The average classification accuracy for the $(4X, adv.)$ group is $98.7\pm0.4\%$. Figure 5 shows a T-SNE embedding of the features of the test images classified by the pretrained model attribution classifier. The figure depicts a similar trend to Figure 3: adversarially-trained models (the square markers) are highly separable, $(4X, L_1)$ models (triangles) less so, and $(2X, L_1)$ (circles) least separable of all.

### 4.1.3 Fingerprinting Unseen SISR Models

So far, we have assumed that the full set of SISR models our attribution classifiers will ever encounter is known and available during training. In real-world applications, such as scraping the web for super-resolved images which make illicit use of a proprietary model, this condition is unlikely to obtain. More likely, attribution classifiers will be met with images from numerous unknown sources, and will need to handle them gracefully. So does our attribution classifier still detect meaningful fingerprints for these unseen models?

To answer this question, we ran the super-resolved images from our 25 pretrained models through the custom model attribution classifier, which has only seen our 180 custom-trained models during training. A T-SNE embedding of the resulting image features is displayed on the right in Figure 5. Notice that these features, taken from the custom attribution classifier, follow a very similar trend to those from the pretrained attribution classifier on the left, which *has* seen these particular SISR models. Class separation is not as good, but the $(4X, adv.)$ models are still modestly separable.

To make this comparison more quantitative, we introduce a metric for cluster similarity: intra- to inter-class distance ratio. We define intra-class distance as the expected distance between two random points in the same class. Inter-class distance is the expected distance from a random point in class $A$ to a random point in class $B$. Formally, the intra- to inter-class distance ratio $R(A, B)$ is defined as:

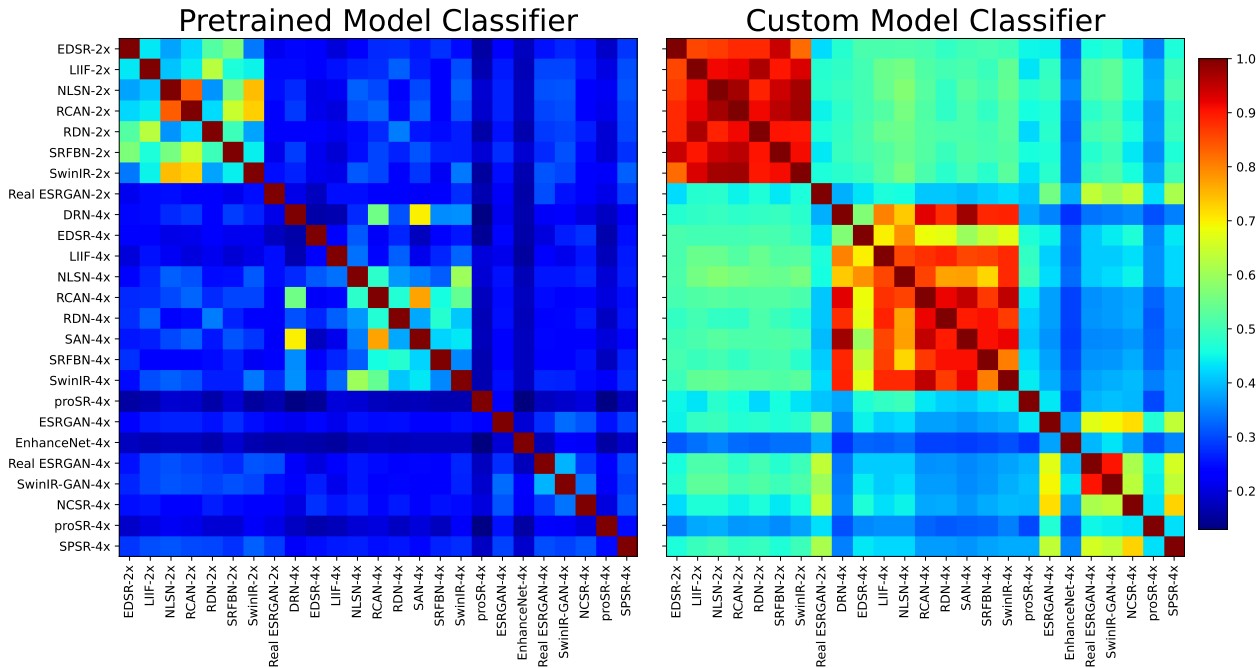

Figure 6: Intra- to inter-class distance ratios for each pair of pretrained SISR models, based on the features encoded by the pretrained vs. custom model classifiers.

$$R(A, B) = \frac{\left(\underset{a_1,a_2 \in A}{\mathbb{E}} \|a_1 - a_2\|_2\right) + \left(\underset{b_1,b_2 \in B}{\mathbb{E}} \|b_1 - b_2\|_2\right)}{2\left(\underset{a \in A, b \in B}{\mathbb{E}} \|a - b\|_2\right)}$$

If A and B are the same, R(A,B)=1. As the clusters become highly separated, R(A,B) approaches zero.

Figure 6 shows this distance ratio for each pair of pretrained models, using the same feature embeddings visualized in Figure 5. This figure shows that class separation is worse across the board when the models were not seen during training. The (4X, adv.) models in the lower-right corner (the last 7 rows of the similarity matrices) are the best-separated from the other models (with the exception of Real-ESRGAN-2x and ProSR-4x). Among these models, average intra- to inter- class distance ratio is around 0.5 on average. In other words, the average distance between two samples in the same class is about half the average distance between two samples from different classes.

## 4.2 Model Parsing

We have demonstrated that small variations in SISR model hyperparameters can lead to distinctive fingerprints. Are these variations in model fingerprints random, or do they contain information about the underlying model hyperparameters? If they contain such information, this could be leveraged towards model parsing. To test this, we train an extensive collection of classifiers, referred to here as *parsers*, to predict each of the experimental hyperparameters of our custom-trained models.

In real-world applications, such parsers would be used to reverse-engineer the hyperparameters of unknown SISR models, which may use architectures, loss functions, etc. which are not in the parser's training data. To simulate this scenario, we pick some hyperparameter value to exclude from each parser's training data. We then evaluate the parser on models with that excluded value. We call this excluded value the *test hyperparameter value*. For example, if a parser's test hyperparameter value is RCAN, this means the parser was trained on the EDSR, RDN, SwinIR, and NLSN models, and tested on the RCAN ones.

Table 5: Test accuracy (%) of 18 model parsers. The "chance baseline" column shows the percent chance of predicting the parameter correctly by random guess. Dashes ("−") indicate parser experiments that would not make sense; *e.g.* withholding all $L_1$ models during training, and then testing the parser's ability to predict the loss function of those $L_1$ models. Rows and columns are ordered so that the parsing problems become more difficult as one moves down or to the right across the table.

| Predicted hyperparam. | Chance baseline | Test hyperparameter value | | | | | |
|---|---|---|---|---|---|---|---|
| | | Seed=3 | Flickr2K | SwinIR | RCAN | VGG+adv. | $L_1$ |
| scale | 50.0 | 100.0±0.0 | 100.0±0.0 | 99.6±0.3 | 99.4±0.0 | 98.9±0.4 | 51.5±0.5 |
| loss | 33.3 | 98.6±0.2 | 98.4±0.2 | 94.3±0.2 | 95.5±0.5 | − | − |
| arch. | 20.0 | 74.2±1.3 | 74.4±2.2 | − | − | 44.3±1.5 | 20.8±0.1 |
| dataset [1] | 25.0 | − | − | 33.0±1.4 | 36.2±1.0 | 31.9±0.3 | 26.1±0.4 |

[1] All models with seeds 2 and 3 were trained with the DIV2K dataset. Therefore, to keep the class frequencies balanced, models with seeds 2 and 3 were excluded during both training and testing of the dataset parsers.

Table 5 shows the test accuracy of 18 model parsers trained and tested on the 180 custom SISR models. The results in this table can be explained as follows: some hyperparameters change the model considerably (scale, choice of $L_1$ or adversarial loss), and some hyperparameters change the model only subtly (dataset and random seed). Parsers perform best when the models in the training and testing sets are quite similar (e.g. they differ only by random seed), and the classes the parser must predict are quite different (e.g. the parser is predicting if the model performs 2X or 4X-scale super resolution).

Performance is worst when the models in the training and testing sets are very different. Figure 2 shows the significant difference in character between images from models trained with $L_1$ loss and those trained adversarially. If we train a parser on the adversarially-trained models and evaluate it on the $L_1$ models, there is a big distributional shift between the training and testing data. This is reflected in the accuracy scores in the $L_1$ column, which hover around the chance baseline. But if we choose to withhold the VGG+adv. models for our test data, our parsers achieve much higher accuracy. They have been trained on models optimized with ResNet+adv. loss, a loss function very similar to the one in the test set. So the test distribution is much closer to the training distribution, and generalization is accordingly better.

These experiments demonstrate the viability of parsing SISR models, but in a limited way. Only some hyperparameters can be reliably parsed, and only when the models are not too far from the training distribution.

### 4.2.1 Pretrained Model Parsing

To study model parsing for pretrained models, we train parsers on all 180 custom-trained SISR models to predict the models' scale, loss, and architecture. (We omit the dataset parser because DIV2K is the only dataset shared by both the custom and pretrained models) We then apply these parsers to images from the 25 pretrained models.

Figure 7 presents a full table of the model parser predictions for each pretrained SISR model. As with the custom model parsers, we find it easy to parse the scale of models with loss functions in the training set. The scale parser we used here was trained on $L_1$, VGG+adv. and ResNet+adv. losses. This parser can predict the scale of our $L_1$-optimized pretrained models with 100% accuracy. EnhanceNet (E.Net) and ProSR are also easy to parse, and are trained with losses very similar to VGG+adv. loss: both leverage a pretrained VGG network for perceptual loss, in combination with adversarial loss. But as shown in Table 5 with test hyperparameter value $L_1$, prediction accuracy for unseen loss functions can be much worse. NCSR and SPSR use very different approaches to super-resolution than the models in the parser's training set, and generalization to these models is accordingly worse. However, ESRGAN also uses a form of VGG loss plus adversarial loss, but our scale parser still performs poorly on it. We consider the exact reason why to be an open question.

Our loss classifier can easily distinguish between $L_1$-optimized and adversarially-trained models: all $L_1$-optimized pretrained models are identified as such with 100% accuracy. The true loss functions of the adversarial SISR models (ESRGAN, EnhanceNet, ProSRGAN, NCSR, SPSR, SwinIR, and Real ESRGAN) are not in the loss classifier's training set. Yet the loss classifier always predicts that these methods were produced with an adversarial loss function (either ResNet+adv. or VGG+adv.).

Architecture prediction is unsuccessful. Our set of pretrained models contains eleven models whose architecture was in the training distribution. Among these models, the architecture classifier can predict the architecture correctly just 22.8% of the time, similar to the random chance accuracy of 20.0%. We speculate that this poor generalization is due to differences in training, such as the learning rate and number of epochs, between custom and pretrained models.

Predicted hyperparameter value

| Actual model | Scale 2x | 4x | Loss $L_1$ | VGG+A. | R.Net+A. | Arch. EDSR | RCAN | RDN | SwinIR | NLSN |
|---|---|---|---|---|---|---|---|---|---|---|
| EDSR-2x | 100 | 0 | 100 | 0 | 0 | 15±6 | 45±7 | 35±8 | 2±1 | 1 |
| LIIF-2x | 100 | 0 | 100 | 0 | 0 | - | - | - | - | - |
| RCAN-2x | 100 | 0 | 100 | 0 | 0 | 2±2 | 63±4 | 30±4 | 2±1 | 1±1 |
| RDN-2x | 100 | 0 | 100 | 0 | 0 | 4±3 | 54±1 | 38±3 | 2±1 | 0 |
| R. ESRGAN-2x | 97 | 2 | 0 | 81±10 | 18±10 | - | - | - | - | - |
| SRFBN-2x | 100 | 0 | 100 | 0 | 0 | - | - | - | - | - |
| SwinIR-2x | 100 | 0 | 100 | 0 | 0 | 2±2 | 68±6 | 21±3 | 6±2 | 2 |
| NLSN-2x | 100 | 0 | 100 | 0 | 0 | 1±1 | 68±3 | 24±4 | 3±1 | 3±1 |
| DRN-4x | 0 | 100 | 100 | 0 | 0 | - | - | - | - | - |
| EDSR-4x | 0 | 100 | 100 | 0 | 0 | 0 | 11±3 | 69±11 | 4±2 | 14±7 |
| LIIF-4x | 0 | 100 | 100 | 0 | 0 | - | - | - | - | - |
| proSR-4x | 0 | 100 | 100 | 0 | 0 | - | - | - | - | - |
| RCAN-4x | 0 | 100 | 100 | 0 | 0 | 0 | 75±6 | 23±7 | 0 | 1 |
| RDN-4x | 0 | 100 | 100 | 0 | 0 | 0 | 62±11 | 36±10 | 0 | 1±1 |
| SAN-4x | 0 | 99 | 100 | 0 | 0 | - | - | - | - | - |
| SRFBN-4x | 0 | 100 | 100 | 0 | 0 | - | - | - | - | - |
| SwinIR-4x | 0 | 99 | 100 | 0 | 0 | 0 | 84±9 | 14±9 | 0 | 1 |
| NLSN-4x | 0 | 100 | 100 | 0 | 0 | 0 | 68±4 | 20±8 | 0 | 10±4 |
| ESRGAN-4x | 51±7 | 49±7 | 0 | 89±4 | 11±4 | - | - | - | - | - |
| E.Net-4x | 0 | 99 | 0 | 1 | 98 | - | - | - | - | - |
| NCSR-4x | 81±9 | 18±9 | 0 | 71±9 | 28±9 | - | - | - | - | - |
| proSR-4x | 5±2 | 94±2 | 0 | 95±3 | 4±3 | - | - | - | - | - |
| R. ESRGAN-4x | 91±3 | 9±3 | 0 | 49±6 | 50±6 | - | - | - | - | - |
| SPSR-4x | 77±7 | 22±7 | 2±2 | 58±8 | 39±6 | - | - | - | - | - |
| SwinIR-adv-4x | 78±8 | 21±8 | 0 | 55±7 | 44±8 | 9±2 | 45±12 | 42±11 | 2±2 | 1 |
| | Scale | | Loss | | | Architecture | | | | |

Figure 7: Parser predictions for the pretrained models. For example, out of the 100 test images for ESRGAN, 51±7 were predicted to come from a model with 2X scale, 49±7 from 4X (values are rounded to integers). Blue boxes inscribe correct predictions. Rows with no blue box have actual values outside the training set. Abbreviations: R. ESRGAN: Real ESRGAN, E.Net: EnhanceNet.

### 4.3 Key Findings

As Yu et al. (2019) showed for GANs, any small change to an SISR model's hyperparameters is sufficient to detect a unique fingerprint: see Section 4.1.1. Our model attribution experiments from Section 4.1 consistently show that model fingerprints are much more distinctive for models trained with adversarial loss than $L_1$ loss, and are slighty more distinctive for 4X models than 2X ones. We posit that this is due to the "open-endedness" of the image synthesis task: the larger the space of valid network outputs, the more opportunity there is for the network to leave distinctive fingerprints. In Section 4.2, our model parsing experiments met with mixed results: our parsers can be either very accurate or very inaccurate, depending on the particular parsing task.

## 5 Conclusion

We have presented the first exploration of model fingerprinting and image attribution specifically focused on single image super-resolution networks. We create a dataset of 205 super-resolution methods. We show that networks trained on more open-ended image synthesis tasks leave more distinctive fingerprints on the images they generate. Our attribution classifiers can learn to detect distinctive fingerprints even for SISR models outside the training set. Our model parsing experiments show that model parsing is possible, but only when the predicted hyperparameter has a large impact on the resulting images, and the model being parsed is similar to those in the training set.

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

# A  Appendix

## A.1  Flickr1K Dataset

Table 6: Main characteristics of the three SR datsets used in this project: Flickr2K (Lim et al., 2017), Div2K (Agustsson & Timofte, 2017), and our "Flickr1K" dataset. Table shows the number of images, pixels per image (ppi), bits per pixel using PNG compression (bpp PNG), and shannon-entropy of the images' greyscale histograms (entropy). For bpp and entropy, we report average (±standard deviation).

| Dataset | Images | Ppi | Bpp PNG | Entropy |
|---|---|---|---|---|
| Flickr2K train | 800 | 2793045 | 12.71($\pm$2.53) | 7.34($\pm$0.57) |
| Flickr2K test | 100 | 2794881 | 12.52($\pm$2.45) | 7.38 ($\pm$0.53) |
| Flickr2K validation | 100 | 2749737 | 12.96($\pm$2.67) | 7.52($\pm$0.29) |
| DIV2K train | 800 | 2779971 | 12.68($\pm$2.79) | 7.48($\pm$0.34) |
| DIV2K validation | 100 | 2733370 | 13.24 ($\pm$2.87) | 7.51($\pm$0.43) |
| Flickr1K train | 100 | 649958 | 12.93 ($\pm$2.75) | 7.30($\pm$0.45) |
| Flickr1K test | 798 | 646188 | 12.89($\pm$2.44) | 7.32($\pm$0.52) |
| Flickr1K validation | 100 | 633369 | 12.39 ($\pm$2.39) | 7.24($\pm$0.54) |

## A.2  SISR Networks Dataset

Our 180 custom-trained models were trained using the BasicSR toolkit (Wang et al., 2018a). Besides the experimental hyperparameters, all training hyperparameters were held constant across the models. For example, all models were trained for 50,000 iterations. This leads to substandard performance of the SwinIR models, which would need to be trained for ten times longer to match the quality of the published models. This poor quality is reflected qualitatively in Figure 2, and quantitatively in Table 8.

### A.2.1 ResNet Loss

One third of our custom-trained super-resolution networks are trained with ResNet + adversarial loss, a combination of discriminator-based adversarial loss, and perceptual loss based on a ResNet18 classifier trained on ImageNet.

Our ResNet-based perceptual loss is computed by passing both the super-resolved image and the ground-truth high-resolution image through the ResNet. We extract the activations output by layer 3 of the ResNet, and take the $L_2$ distance between HR and SR feature vectors as our ResNet loss term. We use this loss term as a drop-in replacement for the VGG-based perceptual loss term used in SRGAN Wang et al. (2018b) and implemented in BasicSR Wang et al. (2018a).

Table 7: Papers which provide the 25 pretrained super-resolution models we use in our dataset (some papers provide multiple models).

| Name | Loss | Scale(s) | |
|------|------|----------|---|
| EDSR Lim et al. (2017) | $L_1$ | DIV2K | 2x, 4x |
| EnhanceNet Sajjadi et al. (2017) | Adv. | MSCOCO | 4x |
| RDN Zhang et al. (2018b) | $L_1$ | DIV2K | 2x, 4x |
| ProSR Wang et al. (2018c) | $L_1$ | DIV2K | 4x |
| ProGanSR Wang et al. (2018c) | Adv. | DIV2K | 4x |
| RCAN Zhang et al. (2018a) | $L_1$ | DIV2K | 2x, 4x |
| ESRGAN Wang et al. (2018b) | Adv. | DIV2K, Flickr2K, OST | 4x |
| SAN Dai et al. (2019) | $L_1$ | DIV2K | 4x |
| SRFBN Li et al. (2019) | $L_1$ | DIV2K | 2x, 4x |
| SPSR Ma et al. (2020) | Adv. | DIV2K | 4x |
| DRN Guo et al. (2020) | $L_1$ | DIV2K | 4x |
| NCSR Kim & Son (2021) | Adv. | DIV2K | 4x |
| SwinIR Liang et al. (2021) | $L_1$, Adv. | DIV2K | 2x, 4x |
| LIIF Chen et al. (2021) | $L_1$ | DIV2K | 2x, 4x |
| Real-ESRGAN Wang et al. (2021) | Adv. | DIV2K | 2x, 4x |
| NLSN Mei et al. (2021) | $L_1$ | DIV2K | 2x, 4x |

Table 8: Full list of all 205 super-resolution models in our dataset, along with their average PSNR and LPIPS scores when compared to the ground truth HR images. Model names are of the form {architecture}-{training dataset}-{scale}-{loss}-{seed}

| SISR model | PSNR ($\pm$ std) | LPIPS ($\pm$ std) |
|------------|------------------|-------------------|
| DRN-div2k-x4-L1-NA-pretrained | 28.31($\pm$6.26) | 0.284($\pm$0.151) |
| EDSR-div2k-x2-L1-NA-pretrained | 32.56($\pm$7.03) | 0.134($\pm$0.115) |
| EDSR-div2k-x2-L1-s1 | 32.27($\pm$6.90) | 0.132($\pm$0.115) |
| EDSR-div2k-x2-L1-s2 | 32.54($\pm$7.06) | 0.133($\pm$0.115) |
| EDSR-div2k-x2-L1-s3 | 32.49($\pm$7.03) | 0.134($\pm$0.115) |
| EDSR-div2k-x2-ResNet+Adv.-s1 | 31.33($\pm$6.60) | 0.074($\pm$0.102) |
| EDSR-div2k-x2-ResNet+Adv.-s2 | 32.25($\pm$7.01) | 0.086($\pm$0.103) |
| EDSR-div2k-x2-ResNet+Adv.-s3 | 31.90($\pm$6.90) | 0.078($\pm$0.102) |
| EDSR-div2k-x2-VGG+Adv.-s1 | 31.91($\pm$6.78) | 0.080($\pm$0.103) |
| EDSR-div2k-x2-VGG+Adv.-s2 | 31.91($\pm$6.79) | 0.080($\pm$0.102) |
| EDSR-div2k-x2-VGG+Adv.-s3 | 32.01($\pm$6.91) | 0.082($\pm$0.102) |
| EDSR-div2k-x4-L1-NA-pretrained | 28.08($\pm$6.26) | 0.293($\pm$0.152) |
| EDSR-div2k-x4-L1-s1 | 27.95($\pm$6.22) | 0.296($\pm$0.151) |
| EDSR-div2k-x4-L1-s2 | 27.95($\pm$6.22) | 0.299($\pm$0.151) |
| EDSR-div2k-x4-L1-s3 | 27.92($\pm$6.22) | 0.307($\pm$0.151) |
| EDSR-div2k-x4-ResNet+Adv.-s1 | 26.96($\pm$5.58) | 0.188($\pm$0.112) |
| EDSR-div2k-x4-ResNet+Adv.-s2 | 26.51($\pm$5.82) | 0.192($\pm$0.107) |

| | | |
|---|---|---|
| EDSR-div2k-x4-ResNet+Adv.-s3 | 25.08($\pm$5.49) | 0.205($\pm$0.115) |
| EDSR-div2k-x4-VGG+Adv.-s1 | 26.57($\pm$5.61) | 0.188($\pm$0.109) |
| EDSR-div2k-x4-VGG+Adv.-s2 | 26.66($\pm$5.43) | 0.185($\pm$0.107) |
| EDSR-div2k-x4-VGG+Adv.-s3 | 26.54($\pm$5.69) | 0.182($\pm$0.105) |
| EDSR-flickr2k-x2-L1-s1 | 32.46($\pm$7.04) | 0.133($\pm$0.115) |
| EDSR-flickr2k-x2-ResNet+Adv.-s1 | 31.98($\pm$6.86) | 0.086($\pm$0.103) |
| EDSR-flickr2k-x2-VGG+Adv.-s1 | 31.38($\pm$6.60) | 0.083($\pm$0.103) |
| EDSR-flickr2k-x4-L1-s1 | 27.90($\pm$6.21) | 0.307($\pm$0.152) |
| EDSR-flickr2k-x4-ResNet+Adv.-s1 | 26.51($\pm$5.65) | 0.195($\pm$0.102) |
| EDSR-flickr2k-x4-VGG+Adv.-s1 | 26.83($\pm$5.64) | 0.207($\pm$0.114) |
| EDSR-quarter div2k-x2-L1-s1 | 32.46($\pm$6.99) | 0.126($\pm$0.113) |
| EDSR-quarter div2k-x2-ResNet+Adv.-s1 | 32.00($\pm$6.95) | 0.075($\pm$0.102) |
| EDSR-quarter div2k-x2-VGG+Adv.-s1 | 32.22($\pm$6.99) | 0.081($\pm$0.103) |
| EDSR-quarter div2k-x4-L1-s1 | 27.94($\pm$6.22) | 0.301($\pm$0.151) |
| EDSR-quarter div2k-x4-ResNet+Adv.-s1 | 26.39($\pm$5.83) | 0.185($\pm$0.106) |
| EDSR-quarter div2k-x4-VGG+Adv.-s1 | 26.45($\pm$5.52) | 0.184($\pm$0.107) |
| EDSR-quarter flickr2k-x2-L1-s1 | 32.45($\pm$7.03) | 0.135($\pm$0.116) |
| EDSR-quarter flickr2k-x2-ResNet+Adv.-s1 | 31.20($\pm$6.31) | 0.093($\pm$0.104) |
| EDSR-quarter flickr2k-x2-VGG+Adv.-s1 | 31.86($\pm$6.79) | 0.089($\pm$0.102) |
| EDSR-quarter flickr2k-x4-L1-s1 | 27.89($\pm$6.18) | 0.305($\pm$0.152) |
| EDSR-quarter flickr2k-x4-ResNet+Adv.-s1 | 26.82($\pm$6.00) | 0.187($\pm$0.105) |
| EDSR-quarter flickr2k-x4-VGG+Adv.-s1 | 27.11($\pm$5.89) | 0.199($\pm$0.112) |
| ESRGAN-NA-x4-ESRGAN-NA-pretrained | 27.48($\pm$6.09) | 0.161($\pm$0.106) |
| EnhanceNet-NA-x4-EnhanceNet-NA-pretrained | 26.48($\pm$5.83) | 0.206($\pm$0.113) |
| LIIF-div2k-x2-L1-NA-pretrained | 32.35($\pm$6.77) | 0.130($\pm$0.115) |
| LIIF-div2k-x4-L1-NA-pretrained | 28.34($\pm$6.34) | 0.283($\pm$0.151) |
| NCSR-div2k-x4-NCSR GAN-NA-pretrained | 27.76($\pm$6.24) | 0.178($\pm$0.112) |
| NLSN-div2k-x2-L1-NA-pretrained | 34.31($\pm$5.20) | 0.103($\pm$0.073) |
| NLSN-div2k-x2-L1-s1 | 32.59($\pm$7.06) | 0.133($\pm$0.115) |
| NLSN-div2k-x2-L1-s2 | 32.22($\pm$6.77) | 0.134($\pm$0.115) |
| NLSN-div2k-x2-L1-s3 | 32.54($\pm$7.06) | 0.130($\pm$0.115) |
| NLSN-div2k-x2-ResNet+Adv.-s1 | 32.08($\pm$6.83) | 0.086($\pm$0.103) |
| NLSN-div2k-x2-ResNet+Adv.-s2 | 31.64($\pm$6.59) | 0.085($\pm$0.102) |
| NLSN-div2k-x2-ResNet+Adv.-s3 | 31.88($\pm$6.70) | 0.090($\pm$0.103) |
| NLSN-div2k-x2-VGG+Adv.-s1 | 31.95($\pm$6.74) | 0.091($\pm$0.103) |
| NLSN-div2k-x2-VGG+Adv.-s2 | 32.00($\pm$6.82) | 0.091($\pm$0.103) |
| NLSN-div2k-x2-VGG+Adv.-s3 | 31.57($\pm$6.54) | 0.095($\pm$0.103) |
| NLSN-div2k-x4-L1-NA-pretrained | 29.51($\pm$5.23) | 0.259($\pm$0.137) |
| NLSN-div2k-x4-L1-s1 | 27.91($\pm$6.03) | 0.289($\pm$0.150) |
| NLSN-div2k-x4-L1-s2 | 28.06($\pm$6.18) | 0.294($\pm$0.150) |
| NLSN-div2k-x4-L1-s3 | 28.15($\pm$6.28) | 0.291($\pm$0.149) |
| NLSN-div2k-x4-ResNet+Adv.-s1 | 26.49($\pm$5.54) | 0.170($\pm$0.105) |
| NLSN-div2k-x4-ResNet+Adv.-s2 | 27.17($\pm$5.82) | 0.193($\pm$0.111) |
| NLSN-div2k-x4-ResNet+Adv.-s3 | 26.53($\pm$5.78) | 0.164($\pm$0.104) |
| NLSN-div2k-x4-VGG+Adv.-s1 | 26.05($\pm$4.92) | 0.180($\pm$0.106) |
| NLSN-div2k-x4-VGG+Adv.-s2 | 26.24($\pm$5.22) | 0.179($\pm$0.108) |
| NLSN-div2k-x4-VGG+Adv.-s3 | 26.56($\pm$5.22) | 0.196($\pm$0.113) |
| NLSN-flickr2k-x2-L1-s1 | 32.29($\pm$6.90) | 0.136($\pm$0.116) |
| NLSN-flickr2k-x2-ResNet+Adv.-s1 | 24.44($\pm$3.75) | 0.117($\pm$0.106) |
| NLSN-flickr2k-x2-VGG+Adv.-s1 | 31.93($\pm$6.78) | 0.095($\pm$0.104) |
| NLSN-flickr2k-x4-L1-s1 | 27.89($\pm$6.05) | 0.291($\pm$0.151) |
| NLSN-flickr2k-x4-ResNet+Adv.-s1 | 26.72($\pm$5.61) | 0.169($\pm$0.108) |
| NLSN-flickr2k-x4-VGG+Adv.-s1 | 25.97($\pm$4.85) | 0.185($\pm$0.110) |
| NLSN-quarter div2k-x2-L1-s1 | 32.50($\pm$6.95) | 0.132($\pm$0.115) |

| | | |
|---|---|---|
| NLSN-quarter div2k-x2-ResNet+Adv.-s1 | 31.62(±6.62) | 0.073(±0.102) |
| NLSN-quarter div2k-x2-VGG+Adv.-s1 | 31.57(±6.51) | 0.090(±0.104) |
| NLSN-quarter div2k-x4-L1-s1 | 28.10(±6.25) | 0.294(±0.151) |
| NLSN-quarter div2k-x4-ResNet+Adv.-s1 | 25.91(±5.27) | 0.175(±0.106) |
| NLSN-quarter div2k-x4-VGG+Adv.-s1 | 26.16(±5.01) | 0.186(±0.106) |
| NLSN-quarter flickr2k-x2-L1-s1 | 32.46(±7.01) | 0.136(±0.116) |
| NLSN-quarter flickr2k-x2-ResNet+Adv.-s1 | 31.30(±6.76) | 0.089(±0.104) |
| NLSN-quarter flickr2k-x2-VGG+Adv.-s1 | 31.32(±6.44) | 0.091(±0.103) |
| NLSN-quarter flickr2k-x4-L1-s1 | 27.87(±6.01) | 0.299(±0.151) |
| NLSN-quarter flickr2k-x4-ResNet+Adv.-s1 | 26.81(±5.62) | 0.172(±0.108) |
| NLSN-quarter flickr2k-x4-VGG+Adv.-s1 | 26.66(±5.42) | 0.189(±0.110) |
| RCAN-div2k-x2-L1-NA-pretrained | 32.79(±7.06) | 0.126(±0.114) |
| RCAN-div2k-x2-L1-s1 | 32.59(±7.07) | 0.135(±0.116) |
| RCAN-div2k-x2-L1-s2 | 32.46(±7.03) | 0.139(±0.117) |
| RCAN-div2k-x2-L1-s3 | 32.67(±7.13) | 0.132(±0.116) |
| RCAN-div2k-x2-ResNet+Adv.-s1 | 31.88(±6.80) | 0.079(±0.102) |
| RCAN-div2k-x2-ResNet+Adv.-s2 | 32.14(±6.95) | 0.080(±0.103) |
| RCAN-div2k-x2-ResNet+Adv.-s3 | 32.07(±6.90) | 0.079(±0.102) |
| RCAN-div2k-x2-VGG+Adv.-s1 | 31.51(±6.58) | 0.091(±0.102) |
| RCAN-div2k-x2-VGG+Adv.-s2 | 31.99(±6.81) | 0.083(±0.102) |
| RCAN-div2k-x2-VGG+Adv.-s3 | 32.06(±6.83) | 0.090(±0.103) |
| RCAN-div2k-x4-L1-NA-pretrained | 28.32(±6.26) | 0.282(±0.152) |
| RCAN-div2k-x4-L1-s1 | 28.17(±6.25) | 0.290(±0.151) |
| RCAN-div2k-x4-L1-s2 | 28.00(±6.23) | 0.302(±0.151) |
| RCAN-div2k-x4-L1-s3 | 28.08(±6.26) | 0.299(±0.152) |
| RCAN-div2k-x4-ResNet+Adv.-s1 | 26.58(±5.44) | 0.174(±0.113) |
| RCAN-div2k-x4-ResNet+Adv.-s2 | 27.41(±6.03) | 0.199(±0.117) |
| RCAN-div2k-x4-ResNet+Adv.-s3 | 27.13(±5.90) | 0.187(±0.099) |
| RCAN-div2k-x4-VGG+Adv.-s1 | 27.33(±5.90) | 0.176(±0.107) |
| RCAN-div2k-x4-VGG+Adv.-s2 | 27.26(±5.85) | 0.176(±0.110) |
| RCAN-div2k-x4-VGG+Adv.-s3 | 27.06(±5.82) | 0.176(±0.106) |
| RCAN-flickr2k-x2-L1-s1 | 32.64(±7.09) | 0.132(±0.116) |
| RCAN-flickr2k-x2-ResNet+Adv.-s1 | 31.87(±6.74) | 0.092(±0.104) |
| RCAN-flickr2k-x2-VGG+Adv.-s1 | 32.13(±6.91) | 0.083(±0.102) |
| RCAN-flickr2k-x4-L1-s1 | 28.06(±6.25) | 0.300(±0.151) |
| RCAN-flickr2k-x4-ResNet+Adv.-s1 | 26.21(±5.30) | 0.168(±0.105) |
| RCAN-flickr2k-x4-VGG+Adv.-s1 | 27.55(±6.08) | 0.167(±0.107) |
| RCAN-quarter div2k-x2-L1-s1 | 32.55(±6.96) | 0.128(±0.116) |
| RCAN-quarter div2k-x2-ResNet+Adv.-s1 | 32.21(±6.90) | 0.083(±0.103) |
| RCAN-quarter div2k-x2-VGG+Adv.-s1 | 32.08(±6.81) | 0.093(±0.104) |
| RCAN-quarter div2k-x4-L1-s1 | 28.15(±6.25) | 0.289(±0.151) |
| RCAN-quarter div2k-x4-ResNet+Adv.-s1 | 27.04(±5.91) | 0.173(±0.101) |
| RCAN-quarter div2k-x4-VGG+Adv.-s1 | 27.38(±5.99) | 0.170(±0.112) |
| RCAN-quarter flickr2k-x2-L1-s1 | 32.55(±7.08) | 0.135(±0.117) |
| RCAN-quarter flickr2k-x2-ResNet+Adv.-s1 | 32.02(±6.89) | 0.090(±0.103) |
| RCAN-quarter flickr2k-x2-VGG+Adv.-s1 | 31.90(±6.79) | 0.091(±0.103) |
| RCAN-quarter flickr2k-x4-L1-s1 | 28.03(±6.25) | 0.302(±0.151) |
| RCAN-quarter flickr2k-x4-ResNet+Adv.-s1 | 27.24(±6.09) | 0.180(±0.105) |
| RCAN-quarter flickr2k-x4-VGG+Adv.-s1 | 27.34(±5.88) | 0.197(±0.118) |
| RDN-div2k-x2-L1-NA-pretrained | 32.34(±6.77) | 0.131(±0.116) |
| RDN-div2k-x2-L1-s1 | 32.47(±6.96) | 0.128(±0.115) |
| RDN-div2k-x2-L1-s2 | 31.75(±6.49) | 0.133(±0.116) |
| RDN-div2k-x2-L1-s3 | 32.51(±7.06) | 0.136(±0.117) |
| RDN-div2k-x2-ResNet+Adv.-s1 | 30.76(±5.94) | 0.088(±0.103) |

| | | |
|---|---|---|
| RDN-div2k-x2-ResNet+Adv.-s2 | 32.19($\pm$6.97) | 0.088($\pm$0.103) |
| RDN-div2k-x2-ResNet+Adv.-s3 | 32.04($\pm$6.81) | 0.081($\pm$0.103) |
| RDN-div2k-x2-VGG+Adv.-s1 | 31.77($\pm$6.55) | 0.079($\pm$0.102) |
| RDN-div2k-x2-VGG+Adv.-s2 | 31.83($\pm$6.58) | 0.089($\pm$0.103) |
| RDN-div2k-x2-VGG+Adv.-s3 | 31.88($\pm$6.70) | 0.087($\pm$0.102) |
| RDN-div2k-x4-L1-NA-pretrained | 28.25($\pm$6.35) | 0.290($\pm$0.152) |
| RDN-div2k-x4-L1-s1 | 28.05($\pm$6.22) | 0.292($\pm$0.150) |
| RDN-div2k-x4-L1-s2 | 28.11($\pm$6.27) | 0.287($\pm$0.152) |
| RDN-div2k-x4-L1-s3 | 27.90($\pm$6.12) | 0.295($\pm$0.152) |
| RDN-div2k-x4-ResNet+Adv.-s1 | 24.41($\pm$4.22) | 0.211($\pm$0.109) |
| RDN-div2k-x4-ResNet+Adv.-s2 | 25.81($\pm$5.52) | 0.184($\pm$0.106) |
| RDN-div2k-x4-ResNet+Adv.-s3 | 26.12($\pm$5.29) | 0.187($\pm$0.102) |
| RDN-div2k-x4-VGG+Adv.-s1 | 26.96($\pm$5.78) | 0.179($\pm$0.109) |
| RDN-div2k-x4-VGG+Adv.-s2 | 27.21($\pm$5.92) | 0.189($\pm$0.108) |
| RDN-div2k-x4-VGG+Adv.-s3 | 26.92($\pm$5.71) | 0.184($\pm$0.109) |
| RDN-flickr2k-x2-L1-s1 | 32.47($\pm$7.02) | 0.136($\pm$0.116) |
| RDN-flickr2k-x2-ResNet+Adv.-s1 | 29.80($\pm$5.63) | 0.092($\pm$0.103) |
| RDN-flickr2k-x2-VGG+Adv.-s1 | 31.82($\pm$6.67) | 0.090($\pm$0.103) |
| RDN-flickr2k-x4-L1-s1 | 28.09($\pm$6.31) | 0.300($\pm$0.151) |
| RDN-flickr2k-x4-ResNet+Adv.-s1 | 26.81($\pm$5.86) | 0.187($\pm$0.104) |
| RDN-flickr2k-x4-VGG+Adv.-s1 | 27.11($\pm$5.91) | 0.186($\pm$0.106) |
| RDN-quarter div2k-x2-L1-s1 | 32.59($\pm$7.11) | 0.136($\pm$0.117) |
| RDN-quarter div2k-x2-ResNet+Adv.-s1 | 31.07($\pm$6.22) | 0.094($\pm$0.104) |
| RDN-quarter div2k-x2-VGG+Adv.-s1 | 32.05($\pm$6.74) | 0.086($\pm$0.103) |
| RDN-quarter div2k-x4-L1-s1 | 28.11($\pm$6.30) | 0.288($\pm$0.149) |
| RDN-quarter div2k-x4-ResNet+Adv.-s1 | 26.52($\pm$5.73) | 0.174($\pm$0.104) |
| RDN-quarter div2k-x4-VGG+Adv.-s1 | 26.95($\pm$5.76) | 0.175($\pm$0.108) |
| RDN-quarter flickr2k-x2-L1-s1 | 32.57($\pm$7.06) | 0.135($\pm$0.117) |
| RDN-quarter flickr2k-x2-ResNet+Adv.-s1 | 31.34($\pm$6.59) | 0.084($\pm$0.102) |
| RDN-quarter flickr2k-x2-VGG+Adv.-s1 | 32.18($\pm$6.94) | 0.091($\pm$0.103) |
| RDN-quarter flickr2k-x4-L1-s1 | 27.99($\pm$6.19) | 0.299($\pm$0.152) |
| RDN-quarter flickr2k-x4-ResNet+Adv.-s1 | 26.30($\pm$5.81) | 0.208($\pm$0.112) |
| RDN-quarter flickr2k-x4-VGG+Adv.-s1 | 27.16($\pm$5.78) | 0.184($\pm$0.110) |
| Real ESRGAN-div2k-x2-GAN-NA-pretrained | 27.79($\pm$6.10) | 0.129($\pm$0.107) |
| Real ESRGAN-div2k-x4-GAN-NA-pretrained | 24.84($\pm$5.53) | 0.206($\pm$0.109) |
| SAN-div2k-x4-L1-NA-pretrained | 28.31($\pm$6.27) | 0.284($\pm$0.152) |
| SPSR-div2k-x4-SPSR GAN-NA-pretrained | 27.44($\pm$6.00) | 0.166($\pm$0.111) |
| SRFBN-NA-x2-L1-NA-pretrained | 32.69($\pm$7.06) | 0.129($\pm$0.115) |
| SRFBN-NA-x4-L1-NA-pretrained | 28.20($\pm$6.25) | 0.292($\pm$0.152) |
| SwinIR-div2k-x2-L1-NA-pretrained | 32.85($\pm$7.06) | 0.124($\pm$0.114) |
| SwinIR-div2k-x2-L1-s1 | 32.23($\pm$6.72) | 0.134($\pm$0.115) |
| SwinIR-div2k-x2-L1-s2 | 32.53($\pm$7.07) | 0.130($\pm$0.114) |
| SwinIR-div2k-x2-L1-s3 | 31.77($\pm$6.43) | 0.129($\pm$0.114) |
| SwinIR-div2k-x2-ResNet+Adv.-s1 | 30.05($\pm$5.86) | 0.093($\pm$0.102) |
| SwinIR-div2k-x2-ResNet+Adv.-s2 | 29.35($\pm$5.16) | 0.097($\pm$0.102) |
| SwinIR-div2k-x2-ResNet+Adv.-s3 | 30.85($\pm$6.19) | 0.083($\pm$0.102) |
| SwinIR-div2k-x2-VGG+Adv.-s1 | 31.60($\pm$6.64) | 0.086($\pm$0.103) |
| SwinIR-div2k-x2-VGG+Adv.-s2 | 31.38($\pm$6.33) | 0.095($\pm$0.104) |
| SwinIR-div2k-x2-VGG+Adv.-s3 | 30.94($\pm$6.08) | 0.092($\pm$0.104) |
| SwinIR-div2k-x4-GAN-NA-pretrained | 24.74($\pm$5.54) | 0.203($\pm$0.108) |
| SwinIR-div2k-x4-L1-NA-pretrained | 28.39($\pm$6.27) | 0.279($\pm$0.152) |
| SwinIR-div2k-x4-L1-s1 | 27.61($\pm$5.74) | 0.295($\pm$0.149) |
| SwinIR-div2k-x4-L1-s2 | 27.98($\pm$6.15) | 0.294($\pm$0.149) |
| SwinIR-div2k-x4-L1-s3 | 27.97($\pm$6.13) | 0.295($\pm$0.149) |

| | | |
|---|---|---|
| SwinIR-div2k-x4-ResNet+Adv.-s1 | 25.45($\pm$5.15) | 0.217($\pm$0.107) |
| SwinIR-div2k-x4-ResNet+Adv.-s2 | 26.30($\pm$5.34) | 0.194($\pm$0.108) |
| SwinIR-div2k-x4-ResNet+Adv.-s3 | 24.20($\pm$4.75) | 0.240($\pm$0.111) |
| SwinIR-div2k-x4-VGG+Adv.-s1 | 26.51($\pm$5.47) | 0.195($\pm$0.106) |
| SwinIR-div2k-x4-VGG+Adv.-s2 | 24.97($\pm$4.45) | 0.216($\pm$0.109) |
| SwinIR-div2k-x4-VGG+Adv.-s3 | 26.39($\pm$5.35) | 0.194($\pm$0.112) |
| SwinIR-flickr2k-x2-L1-s1 | 32.01($\pm$6.71) | 0.132($\pm$0.115) |
| SwinIR-flickr2k-x2-ResNet+Adv.-s1 | 31.03($\pm$6.37) | 0.094($\pm$0.104) |
| SwinIR-flickr2k-x2-VGG+Adv.-s1 | 31.14($\pm$6.28) | 0.090($\pm$0.104) |
| SwinIR-flickr2k-x4-L1-s1 | 28.13($\pm$6.18) | 0.286($\pm$0.149) |
| SwinIR-flickr2k-x4-ResNet+Adv.-s1 | 25.88($\pm$5.15) | 0.228($\pm$0.103) |
| SwinIR-flickr2k-x4-VGG+Adv.-s1 | 24.99($\pm$4.28) | 0.215($\pm$0.109) |
| SwinIR-quarter div2k-x2-L1-s1 | 32.26($\pm$6.79) | 0.133($\pm$0.115) |
| SwinIR-quarter div2k-x2-ResNet+Adv.-s1 | 31.43($\pm$6.63) | 0.087($\pm$0.102) |
| SwinIR-quarter div2k-x2-VGG+Adv.-s1 | 31.52($\pm$6.59) | 0.088($\pm$0.103) |
| SwinIR-quarter div2k-x4-L1-s1 | 27.94($\pm$6.09) | 0.285($\pm$0.148) |
| SwinIR-quarter div2k-x4-ResNet+Adv.-s1 | 25.67($\pm$5.07) | 0.217($\pm$0.107) |
| SwinIR-quarter div2k-x4-VGG+Adv.-s1 | 26.10($\pm$5.19) | 0.201($\pm$0.109) |
| SwinIR-quarter flickr2k-x2-L1-s1 | 32.52($\pm$7.00) | 0.132($\pm$0.115) |
| SwinIR-quarter flickr2k-x2-ResNet+Adv.-s1 | 28.88($\pm$5.04) | 0.081($\pm$0.101) |
| SwinIR-quarter flickr2k-x2-VGG+Adv.-s1 | 31.56($\pm$6.52) | 0.096($\pm$0.105) |
| SwinIR-quarter flickr2k-x4-L1-s1 | 27.97($\pm$6.15) | 0.296($\pm$0.150) |
| SwinIR-quarter flickr2k-x4-ResNet+Adv.-s1 | 25.38($\pm$4.89) | 0.219($\pm$0.108) |
| SwinIR-quarter flickr2k-x4-VGG+Adv.-s1 | 26.06($\pm$5.00) | 0.222($\pm$0.118) |
| proSR-div2k-x4-L1-NA-pretrained | 28.16($\pm$6.20) | 0.292($\pm$0.151) |
| proSR-div2k-x4-ProSRGAN-NA-pretrained | 27.58($\pm$6.17) | 0.167($\pm$0.109) |

