# OpenReview forum: "Fingerprints of Super Resolution Networks"
_TMLR — Accepted by TMLR_

### Review · Reviewer_JV1y · 2022-07-04

**Summary Of Contributions:**

This paper studies the fingerprinting problem (including deepfake detection, model attribution, model parsing, etc.) The paper focused on the single-image super-resolution task and considered a wide choice of model architecture, dataset, scale, loss and seed. The paper studied the impact of different hyperparameters on fingerprinting.


**Broader Impact Concerns:**

It may be better to discuss whether fingerprinting will pose security concerns to applications with generative models.


**Requested Changes:**

*  Consider the effect of methods that try to bypass detection in model training or inference.
* Consider the effect of hyperparameters of the detection models, in addition to the hyperparameters of the generative models.
* Consider a few different generative tasks.


**Strengths And Weaknesses:**

Strengths:

* The paper constructed a dataset for studying the fingerprinting problem on the single-image super-resolution task, which may help future research if the dataset is made public.

* The paper compared the effect of different hyperparameters of the target models (single-image super-resolution models) on fingerprinting.

Weaknesses:

* [Major] Considered factors are hyperparameters, which are relatively minor factors. In addition to the hyperparameters of the generative models, the hyperpameters of the detection models also need to be considered, which may also affect results.

* [Major] I think it is important to study the effect of methods that try to bypass detection in model training or inference (if applicable). For example, a generative network may try to perturb its output to increase the difficulty of detection. Such factors are directly related to fingerprinting.

* [Minor] Only single-image super-resolution models are considered. It will be better to consider a few different generative tasks.

---

> ### Author Response · Authors · 2022-08-20
> **Author Response**
>
> Thank you for the insightful review. Below are responses to individual points:
>
> - **“...which may help future research if the dataset is made public”**
>   - We certainly hope so! We will release our data and code upon publication of the paper.
>
> - **“Consider the effect of methods that try to bypass detection in model training or inference.”**
>   - This is certainly an interesting research direction, which others have investigated for other kinds of generative models, e.g. [1]. However, as we are the first to study model attribution and parsing specifically focused on networks which perform low-level image enhancements (SISR networks), we feel it is sufficient to establish some basic facts about this new problem domain, and leave more sophisticated experiments (e.g. adversarial scenarios) for future work.
>
> - **“Consider the effect of hyperparameters of the detection models, in addition to the hyperparameters of the generative models.”**
>   - We agree that the hyperparameters of the detection models will probably have an impact on their effectiveness. But our paper is primarily focused on introducing a new set of classification problems (SISR attribution and parsing) and establishing a reasonable baseline for how difficult these problems are. To establish a credible baseline, we limited ourselves to using standard image classification methods. Among the four modern classification architectures we tried, ConvNext (published in CVPR 2022) performed the best (see Table 2), so this is what we used throughout the paper. More finely tuned detection models may surpass our results. However, it is beyond the scope of our study, and we will explore other attribution and parsing models in our future work.
>
> - **“Consider a few different generative tasks.”**
>   - The space of potential generative models is vast, and we only have the resources to explore a small sample of it. When designing the experiments we conducted in this paper, we faced many tradeoffs of breadth vs. depth: do we compare the fingerprints of a broad range of models, or focus on the differences among a much narrower range of models? We chose to stay very tightly focused on a single kind of generative task: Single-image super-resolution. By focusing on training a large number of these networks, many of which are similar to each other, we get a much more comprehensive picture of attribution and parsing in this space, and are able to get a deeper understanding of the problem. There are findings we expect to carry over to other generative models, like how different loss functions lead to distinctive fingerprints, or how more “open-ended” image synthesis problems leave more room to embed fingerprints. Empirically confirming that these findings carry over as we expect is a good direction for future work.
>
> - **"It may be better to discuss whether fingerprinting will pose security concerns to applications with generative models."**
>   - This is an interesting idea. I could imagine this being a security concern if an application/organization wished to be secretive about which generative model they are using, or about when they update to new versions of their model, etc. Model parsing threatens to reverse-engineer design decisions that an organization might wish to keep secret, although the parsers trained in this paper are probably not effective enough to pose a serious threat yet. But future improvement on the parsers could indeed pose a potential security concern.
>
> **References:**
> 1. Carlini, Nicholas, and Hany Farid. "Evading deepfake-image detectors with white-and black-box attacks." *Proceedings of the IEEE/CVF conference on computer vision and pattern recognition workshops.* 2020.

---

> ### Comment · Reviewer_JV1y · 2022-08-24
> **Follow-up**
>
> Thanks to the authors for the response and revision.
>
> >In a concurrent work, Asnani et al. (2021) train a convolutional network to extract a fingerprint from a
> generated image, and to predict the hyperparameters of various image generation models from this fingerprint
> . Their method, like ours, can be used for attribution and model parsing. Their study covers a diverse domain
> of 100 image generation models, including GANs, variational autoencoders, and one SISR network: ESRGAN
> (Wang et al., 2018b). Our work, by contrast, is focused specifically on SISR models, which generate images
> very close to a ground truth and likely leave less flexibility to embed fingerprints. We are the first to attempt
> to reverse-engineer hyperparameters of SISR models from the images they generate.
>
> This sounds like that the problem studied in this paper (on SISR) has been covered in Asnani et al. (2021) which sounds more general than this work? So the contribution of this paper over existing works is not quite clear and there is no detailed comparison. And I doubt if the claim "We are the first to attempt to reverse-engineer hyperparameters of SISR models from the images they generate" is true. The abstract of Asnani et al. (2021) mentioned "We propose to perform reverse engineering of GMs to infer model hyperparameters from the images generated by these models" which sounds to have covered the case in this paper.

---

> > ### Author Response · Authors · 2022-08-25
> > **Author Response to follow-up**
> >
> > Thank you very much for the quick feedback!
> >
> > Asnani et al. (2021) analyze a dataset of images collected from 116 different generative models, only two of which we identified as super-resolution methods: ESRGAN and SRFLOW. Only one of these models, SRFLOW, made it into one of their four test sets (see Table 1 from their supplementary material). There is little analysis in the paper that specifically focuses on these super resolution models; they are treated as a couple data points in a much larger dataset. This is why we felt justified in claiming to be the first to study reverse-engineering SISR models. But I see your point, as also described in our paper, Asnani et al. (2021) did include SISR models into their study and therefore, as written, our original claim is not sufficiently precise. We will update our claim to be more accurate.
> >
> > We agree that Asnani et al. (2021) covered the case of SISR of this paper. But as discussed above, they did not specifically focus on SISR, which poses unique challenges for model parsing. Compared to many GAN based image generating neural networks, the results of SISR models are now very close to the ground truth, which provides less flexibility to embed / hide a signature in the results and images generated by different SISR models are very similar to each other, which could potentially make it more difficult for model parsing. This is the unique problem our paper investigates.

---

### Review · Reviewer_dd7R · 2022-07-18

**Summary Of Contributions:**

This paper studies the problem of fingerprinting SISR models, making hypotesis about the fingerprints and test them with varios current SISR models with diffrent setups.
Using those models, the paper create a SR dataset.
It test diffrent attribution models and conclude the use of pre-trained model, as well as test over OOD models (i.e SISR models the classification model didn't train with their data).



**Broader Impact Concerns:**

I think that this paper asks very interesting questions about SISR models, and would like to see future research about it that might mitigate newer models with some general framework. also, the dataset should be published with the paper since it can be used for future research on the topic.

**Requested Changes:**

You have twice in the paper two consecutive "the" ("the the"); please erase those.

In section 4.2: I suggest you don't explain table 5 in such detail (the paragraph about moving down and left to the easier task, that's also by convention usually easier to harder ordered by the left to right), it seems easier to just look at the table, but if you want to guild the reader more, I suggest it would be part of the table description above it.

Figure 6: predicting the scale looks too easy if you have so many 100% acc. is the prediction between all SISR models? if so, I'm not sure I understand what sums to 100, but if you predict each parameter separately it makes more sense, but then it does not support the claim that the models with 4x have a larger fingerprint if you exclude predicting the scale and the loss.

Appendix of Classifier Training Procedure: I think you supplemented all the data in the paper; besides maybe the optimizer, I don't think you need this appendix.

Loss: I think you should explicitly write the perceptual loss (VGG loss etc..).

You state that you made a dataset from querying flicker; there is room to elaborate if you used different artists, specifically different camera sets, to avoid data leakage.

I don't feel like the conclusion about fingerprints is very decisive; I also don't see strong statistical evidence for the fingerprint; please add to the conclusion what you learned about your hypothesis that you mentioned in this paper.


**Strengths And Weaknesses:**

Strengths:
* novelty of studying the fingerprint of SISR models exclusively.
* very extensive study, using a variety of models and data.
* provide a lot of data about the experiments

Weaknesses:
* The authors' hypothesis does not have strong statistical evidence besides scaling and loss (in most cases).
 you can dispute that comment, but you also ran each experiment once, as with different data/different seeds you might be able to support your claims better.
* The current setup and dataset make it limited to add newer models in the future (this is semi-weakness since it is not the goal of the paper)

*Post-rebuttal review:*

I have carefully reviewed the revised version of the manuscript taking into account my and other reviewers' feedback and accordingly authors' responses. I believe that the revised version is substantially improved and the current version of the manuscript is in solid shape and will be beneficial to the community. Hence I vote for its acceptance.

---

> ### Author Response · Authors · 2022-08-20
> **Author Response**
>
> We thank the reviewer for their suggestions, which have helped to improve our paper. Below are responses to the reviewer’s individual points:
>
> - **“The hypotesis the authors makes, does not have a strong statistical avidance, besides of scaling and loss (in most cases). you can dispute that commant, but you also ran each experiment once, as with diffrent data/ diffrent seeds you might be able to support your claims better.”**
>   - We agree that running the experiments with multiple random seeds makes our paper stronger. Unfortunately, due to the slowness of training these classifiers, we could not retrain all of the classifiers used in this paper multiple times for different random seeds or datasets within the two-week rebuttal period.
>
> - **“You have twice in the paper two consecutive "the" ("the the"), please erase those.”**
>   - Thanks for catching this! We have fixed these problems.
>
> - **“In section 4.2: i suggest you don't explain table 5 in such details (the paragraph about moving down and left to the easier task, that's also by convention usually easier to harder ordered by left to right), it seems easier to just look at the table, but if you want to guild the reader more, i suggest it would be part of the table description above it.”**
>   - As you suggested, we have reordered the columns of the table, and rewritten the caption to contain some information that was in the main text.
>
> - **“Figure 6: predicting the scale looks too easy if you have so many 100% acc. is the prediction is between all SISR models? if so, i'm not sure i understand what sums to 100, but if you predict each parameter seperatyly it make more sence, but then it does not support the claim that the models with 4x have a larger fingerprint, if you exclude predicting the scale and the loss.”**
>   - *(Note: we have added a new figure in the revised version. Somewhat confusingly, the figure that the reviewer refers to as Figure 6 is now Figure 7)* Your second guess is correct, we predict each parameter separately. But you are correct that this experiment does not support our claim that 4x models have a larger fingerprint. Many of our attribution experiments *do* support this claim, see Tables 3 and 4, and Figures 3, 4, 5, and 6.  But for the experimental results shown in Figure 7, super-resolution scale is just not an important explanatory factor. The results in this table are better explained by other considerations, which we discuss in Section 4.2.1.
>
> - **“Appendix of Classifier Training Procedure: i think you suplemented all the data in the paper, besides maybe of the optimizer, i don't think you need this apendix.”**
>   - We followed the suggestion and have moved this information up into the main body of the paper and removed this section of the appendix.
>
> - **“Loss: I think you should explicitly write the perceptual loss (VGG loss etc..).”**
>   - We followed the suggestion and added that in Section 3.1.
> - **“You state that you made a dataset from quering fliker, there is room to elaborate if you used diffrent artist, specifically diffrent camera sets, to avoid data leakage.”**
>   - These 1,000 images come from 502 different Flickr accounts and 312 different camera models (as collected from the photos’ EXIF data). We have added some information about the diversity of photographers and cameras in our Flickr1K dataset. Regarding data leakage: due to the way we construct our dataset, image-related variables like artist, camera model, etc. are guaranteed to be uncorrelated with our classifier labels. Our dataset contains 205 copies of each original Flickr image, one for each of our 205 super-resolution models. So there is a version of each image for each possible classifier label.
>
> - **“I don't feel like the conclution about the fingerprints is very decicive, i also don't see strong statisticall avidance for the fingerprint, please add to the conclution about what you learn about your hypotesis that you mentioned in this paper.”**
>   - Thank you for pointing out this concern and suggestion. In the new version of our manuscript, we have added a “key findings” section just before the conclusion. This subsection references specific sections in the paper which support our conclusions.
>
> Thanks again for all the feedback!

---

### Review · Reviewer_KmiB · 2022-08-06

**Summary Of Contributions:**

This paper studied the problem of model parsing/model fingerprint identification for single image super-resolution (SISR) networks. I find this problem formulation interesting and agree that this research area is under-explored. The collection of SISR models and the extensive study of attribution classification are plausible. I am generally optimistic about this submission.

**Requested Changes:**

Please address the questions listed in weaknesses.

**Strengths And Weaknesses:**

Strengths:

+ Model parsing/reverse engineering of model fingerprints is an exciting and novel research problem.

+ The collection of SISR models would benefit future studies on reverse engineering of model fingerprints.

+ The empirical study is extensive. Fig. 6 is insightful for the model parser.

Weaknesses:

- It is not clear to me about the novelty of the classification network part.

- I would like to see a more detailed analysis of the choice of model architecture, super-resolution scale factor, loss function, training dataset, and random seed. What is the rationale behind focusing on these hyperparameters and how they are set in experiments.

---

> ### Author Response · Authors · 2022-08-20
> **Author Response**
>
> Thank you for your helpful feedback and suggestion, which has improved the clarity of our paper. Below are responses to the reviewer’s individual points:
>
> - **“It is not clear to me about the novelty of the classification network part.”**
>   - The novelty of our paper is in the novelty of the classification problems we study, we do not claim any novelty in our classification network. In fact, our choice of classification network and training procedure is deliberately very standard. Our paper establishes a simple baseline for how well modern classification networks can do on these attribution and parsing problems. That being said, our paper examined several other recent classification networks: XceptionNet, EfficientNet, and ResNet50, which did not perform as well as ConvNext (see Table 2 in the paper).
> - **“I would like to see a more detailed analysis of the choice of model architecture, super-resolution scale factor, loss function, training dataset, and random seed. What is the rationale behind focusing on these hyperparameters and how they are set in experiments.”**
>   - We see model architecture, scale factor, loss function, and training dataset as the key factors of a standard SISR model. In fact, the only SISR model hyperparameters that don’t fall into one of these categories are related to the optimization process (learning rate, batch size, etc.). We added the fifth experimental hyperparameter, random training seed, as a kind of control group, under the assumption that changing any meaningful parameter of the SISR model will have an impact at least as significant as changing the random seed. The particular values we set these hyperparameters to are simply popular values from the SISR literature; popular architectures, training datasets, etc. We have added this discussion of why we chose these parameters to Section 3.1 Paragraph 2.

---

> > ### Comment · Reviewer_KmiB · 2022-09-03
> > **Thanks for the response**
> >
> > Thank authors for addressing my comments. Although the technical novelty of this paper is a bit limited, the studied problem is interesting, and the insights to reverse engineering of black-box neural networks are plausible. Thus, I tend to accept it.

---

### Review · Reviewer_Kicq · 2022-08-07

**Summary Of Contributions:**

The paper studied the problem of fingerprints of super resolution networks. Specifically, it crafted a benchmark dataset from 205 SISR models and conducted extensive experiments on those models to evaluate fingerprints for model attribution and model parsing. Results have shown some interesting observations such as the generalization of model attribution classifier, and the potential reverse-engineering of a black box model.

**Broader Impact Concerns:**

N/A.

**Requested Changes:**

- A comparison of baseline methods is necessary and can make the paper more convincing (if applicable).
- I hope that authors can elaborate more on the potential practical use of the conclusions in the paper to improve performance of the SISR models.
- Experiments can be further improved, such as including some statistical results of the generalization performance of model attribution classifiers rather than visualization.

**Strengths And Weaknesses:**

Strengths:
- The proposed dataset of generated images from a large number of SISR models can benefit future research.
- The experiments are extensive and the analysis of empirical results provides insights for two primary tasks in the paper, model attribution and model parsing.

Weaknesses:
- The paper did not include comparison with baseline methods and it was not clear whether the proposed classifier model can be compared with methods in model attribution and model parsing respectively.
- I am curious whether the analysis of fingerprints in model attribution and model parsing can be further extended to improving performance of the original SISR models and that would be more exciting.
- The generalization of model attribution classifiers was only concluded from the embedding visualization in Figure 5. A more rigorous study is necessary to come to the conclusion.

---

> ### Author Response · Authors · 2022-08-20
> **Author Response**
>
> Thank you for your review. It has helped us improve the clarity of our paper. Below are responses to the reviewer’s individual points:
>
> - **“A comparison of baseline methods is necessary and can make the paper more convincing (if applicable).”**
>   - In Table 4, we present a brief comparison of our model attribution classifier with the PRNU-based fingerprinting method from [1]. ConvNext-based classification significantly outperforms that simple baseline on this attribution task. We also compare the performance of the ConvNext architecture on the 205-way model attribution problem to the XceptionNet and Resnet50 architectures, which were used for cnn-genreated image detection in [2] and [3] respectively. Convnext Performed the best on this attribution task (see Table 2).
>
> - **“I hope that authors can elaborate more on the potential practical use of the conclusions in the paper to improve performance of the SISR models.”**
>   - We thank the reviewer for the suggestion; we agree that this is an interesting potential research direction. However, our experiments are not directly concerned with improving the performance of SISR models. We are primarily interested in studying model attribution and parsing problems on a specific domain of image generation models (SISR models) which haven’t received focused attention yet. One potential practical use case is image forensics: our experiments suggest that it is possible to determine which neural network modified an image, even when that neural network only performs low-level image enhancements.
>
> - **“Experiments can be further improved, such as including some statistical results of the generalization performance of model attribution classifiers rather than visualization.”**
>   - At the reviewer’s suggestion, we have included some quantitative results to back up the qualitative judgements made about this figure. See the new Figure 6 and corresponding discussion.
>
> **References**
> 1. Francesco Marra, Diego Gragnaniello, Luisa Verdoliva, and Giovanni Poggi. Do gans leave artificial fingerprints? In *IEEE Conference on Multimedia Information Processing and Retrieval,* 2019.
> 2. Andreas Rössler, Davide Cozzolino, Luisa Verdoliva, Christian Riess, Justus Thies, and Matthias Nießner. FaceForensics++: Learning to detect manipulated facial images. In *IEEE International Conference on Computer Vision,* 2019.
> 3. Sheng-Yu Wang, Oliver Wang, Richard Zhang, Andrew Owens, and Alexei A Efros. Cnn-generated images are surprisingly easy to spot...for now. In *IEEE conference on computer vision and pattern recognition,* 2020.

---

### Review · Reviewer_oBKa · 2022-08-09

**Summary Of Contributions:**

This paper explores the fingerprints of single-image super-resolution networks. The author generates super-resolved images by 205 different models with diverse hyper-parameter (e.g. scale, loss, etc.). With these datasets, the author demonstrates attribution classifier is able to discriminate the attribute from the image itself. Moreover, it is possible to reverse-engineer some of the model's hyperparameters from the images.

**Requested Changes:**

Please address the questions listed in weaknesses.

**Strengths And Weaknesses:**

Strength
* Interesting observation on the fingerprint of SISR
* Construction of dataset of SISR networks
* Extensive experimental fingerprint results on diverse hyper-parameter of SISR

Weakness
* I think it is difficult to find the strong contribution of this paper. The existence of fingerprints is already known and shown in Yu et al (2019). It is difficult to say that it is a large contribution to experimentally confirming the fingerprint characteristics in the SISR model rather than the GAN model.
* Moreover, it is difficult to know what is a big difference in fingerprints between SISR and GAN.
* Presentation is poor in Figure 5. It is difficult to distinguish the dot. And is there any meaning of shape?
* Statical results should be included. I hope the author conducts at least 3 random seeds on the classifier.

Question
* I do not quite understand why "-" indicates no sense experiments in table 5.
* I might miss, how did you split the train set and test set for the attribution classifier?

---

> ### Author Response · Authors · 2022-08-20
> **Author Response**
>
> Thank you for your review. It has helped us improve the clarity of our paper. Below are responses to the reviewer’s individual points:
>
> - **“I think it is difficult to find the strong contribution of this paper. The existence of fingerprints is already known and shown in Yu et al (2019). It is difficult to say that it is a large contribution to experimentally confirming the fingerprint characteristics in the SISR model rather than the GAN model.”**
>
>   - We think that our paper contributes more than just a confirmation of fingerprint characteristics for SISR models. By testing for fingerprint characteristics on several kinds of SISR models, we can begin to tease apart what makes GAN fingerprints unique: is it the adversarial training procedure? Or the “open-endedness” of the generative problem? Our experiments offer some insight into how unique these fingerprints are under various conditions. We show that they are more distinctive for 4X models than 2X models, and more distinctive when trained adversarially than when optimized for L1 loss. We hypothesize that this is due to the increased “open-endedness” of the synthesis task. We suspect that this open-endedness hypothesis will be predictive of model fingerprint uniqueness in other settings as well.
>   We also show that some model hyperparameters can be “parsed” by looking at the generated images, but only when the generative models are close to those in the parser’s training distribution. We suspect that this is also a general principle which will apply when attempting to parse the hyperparameters of other generative models.
>   Another contribution we hope the research community will find valuable is our dataset of 205 SISR models and 205,000 super-resolved images, which we will release when the paper is published.
>
> - **“Moreover, it is difficult to know what is a big difference in fingerprints between SISR and GAN.”**
>   - Since we do not conduct experiments on traditional GANs, we agree that our paper does not make any precise claims about the difference between the fingerprints of SISR models and more traditional GANs. Instead, our experiments simply confirm that SISR models *also* have distinctive fingerprints. Our experiments also draw comparisons between the fingerprints of different SISR models. We show that the SISR models which are closer to traditional, unconditional GANs are relatively easier to uniquely identify: for example, our adversarially trained 4X SISR models can be attributed with 99.8% accuracy, while our 2x, L1-optimized models can only be attributed with 88.6% accuracy.
>
> - **“Presentation is poor in Figure 5. It is difficult to distinguish the dot. And is there any meaning of shape?”**
>   - Thanks for pointing out these issues. In the new revision, we have altered Figure 5 to be larger, making the markers more distinct. (where the markers are tightly clustered together, we don’t think it’s important to our conclusion to be able to distinguish between individual markers, as long as one can tell the shape and color of the marker being used.) And yes, the shape is a function of the scale and loss function of the SISR method. This information has been added to the legend.
>
> - **“Statistical results should be included. I hope the author conducts at least 3 random seeds on the classifier.”**
>   - We agree that running the experiments with at least 3 random seeds would make our paper stronger. However, unfortunately, due to the slowness of training these classifiers, we could not retrain all of the classifiers used in this paper for additional two new random seeds within the two-week rebuttal period.
>
> - **“I do not quite understand why "-" indicates no sense experiments in table 5.”**
>   - “-” indicates a train/test split of the SISR models which would split them by their class label. For example, say the parser is trying to predict the model’s loss function, and all the models with loss=L1 are withheld as testing data. This is like splitting the CIFAR10 dataset by putting all the images of frogs in the test set, and everything else in the training set. We can expect accuracy on the test set to be near 0, since none of the images in its training set were labeled as frogs.
>
> - **“I might miss, how did you split the train set and test set for the attribution classifier?”**
>   - We discuss this in Section 3.3, Classification Networks: “Our classifiers are trained with our super-resolved image dataset on just 800 images from each SISR model. We reserve an additional 100 images for validation, and 100 for testing.“
> An important note here is that we use the same 800 images for training, 100 for testing, and 100 for validation across all SISR models, thus guaranteeing that the high-level image content is perfectly uncorrelated with the SISR model, and that the model is validated and tested on Flickr images it has never seen during training.

---

### Author Response · Authors · 2022-08-20
**Revision of our Paper**

We thank the reviewers for their thoughtful comments and suggestions. Based on the reviewers’ suggestions, we have uploaded a revised version of the paper. To make this revision easier to review, newly added text is colored blue, and there is also a new figure (Figure 6). All responses to reviewers’ specific points will be in replies to individual reviews.

---

### Decision · Action_Editors · 2022-09-04

**Recommendation:** Accept with minor revision

**Comment:**

This paper studies fingerprints of single-image super-resolution networks. The author investigates 205 different SISR models with a number of hyper-parameter settings. On a created dataset, the paper demonstrates that 1) the choice of scaling factor and loss function significantly impacts model distinctiveness; 2) the fingerprints of SISR models trained with an adversarial loss are highly sensitive to small changes in hyperparameters; 3) the SISR model attribution classifier generalizes to models outside the training set; 4) it is possible to reverse-engineer some of a model’s hyperparameters from the output images.

This is a purely empirical study paper, which tries to study the fingerprint property of SISR models. Due to multiple rounds of reviewers invitations, the reviewing time got a little delayed, with totally 5 reviewers involved. All reviewers agree the problem is interesting, and the created dataset could benefit the research community. Some major concerns from the reviewers are:

1. The contribution does not seem significant, given that fingerprints in GAN are also studied. I agree on this. As SISR can be in a smaller domain of GAN, the current studies and discoveries do not seem to be broadly useful. For improvement, the authors are encouraged to consider some potential applications of the findings, e.g., image forensics as mentioned by the authors.
2. The conclusion is not statistically verified. Since this is an empirical study, to claim some conclusions, it is necessary to do it statistically, e.g., by multiple runs.

Overall, the problem of fingerprints in SISR is interesting to investigate. Based on the two acceptance standards: 1) I believe some individuals in TMLR's audience will be interested in the findings of this paper; and 2) all the claims made in the submission have not been well supported by accurate, convincing and clear evidence. Some potential suggestions for improvement: conduct multiple runs of the models tp achieved statistical verification, as suggested by the reviewers; encouraged but not necessary: extend the paper with broader factors, for example, investigating if it is possible to break the fingerprints via some adversarial robustness methods such as adversarial training.

---

> ### Author Response · Authors · 2022-10-04
> **Author Response**
>
> Thank you for your recommendations. We have uploaded a camera-ready revision which incorporates the reviewers' suggestion to statistically verify our results by repeating multiple runs of our classification experiments with different random seeds. We have also added a GitHub link to the code for this project. We will add our dataset along with instructions on how to reproduce our experiments to that GitHub repository very soon.